# Expression of USP25 associates with fibrosis, inflammation and metabolism changes in IgG4-related disease

Panpan Jiang[1], Yukai Jing[1], Siyu Zhao[2], Caini Lan[1], Lu Yang[1], Xin Dai[1], Li Luo[1], Shaozhe Cai[3], Yingzi Zhu[3], Heather Miller[4], Juan Lai[5], Xin Zhang [5], Xiaochao Zhao[5], Yonggui Wu[6], Jingzhi Yang[7], Wen Zhang[8], Fei Guan[1], Bo Zhong [9,10], Hisanori Umehara [11], Jiahui Lei[1], Lingli Dong [3] ✉ & Chaohong Liu [1] ✉

IgG4-related disease (IgG4-RD) has complex clinical manifestations ranging from fibrosis and inflammation to deregulated metabolism. The molecular mechanisms underpinning these phenotypes are unclear. In this study, by using IgG4-RD patient peripheral blood mononuclear cells (PBMCs), IgG4-RD cell lines and *Usp*25 knockout mice, we show that ubiquitin-specific protease 25 (USP25) engages in multiple pathways to regulate fibrotic and inflammatory pathways that are characteristic to IgG4-RD. Reduced USP25 expression in IgG4-RD leads to increased SMAD3 activation, which contributes to fibrosis and induces inflammation through the IL-1β inflammatory axis. Mechanistically, USP25 prevents ubiquitination of RAC1, thus, downregulation of USP25 leads to ubiquitination and degradation of RAC1. Decreased RAC1 levels result in reduced aldolase A release from the actin cytoskeleton, which then lowers glycolysis. The expression of LYN, a component of the B cell receptor signalosome is also reduced in USP25-deficient B cells, which might result in B cell activation deficiency. Altogether, our results indicate a potential anti-inflammatory and anti-fibrotic role for USP25 and make USP25 a promising diagnostic marker and potential therapeutic target in IgG4-RD.

IgG4-related disease (IgG4-RD) is a systemic fibroinflammatory process characterized by a large number of IgG4 producing plasma cells infiltrating target organs[1]. Numerous studies have shown that IgG4-RD leads to multiple inflammatory diseases, including autoimmune pancreatitis, hypophysitis, Mikulicz's disease, interstitial pneumonitis, and interstitial nephritis[2]. Fibrosis develops during periods of chronic inflammation and recent studies have revealed that B cells act as a key player in tissue fibrosis in IgG4-RD[3]. B cells produce the pro-fibrotic molecule, which stimulates fibroblasts to produce collagen and enzymes related to extracellular matrix remodeling[4]. However, the underlying mechanism of how B cells promote inflammation and fibrosis in IgG4-RD is unclear.

Immunodysregulation of specific B cell subsets is key to the pathogenesis of IgG4-RD[5,6]. It was reported that the differentiation of specific B cells in IgG4-RD patients is altered[7–10]. Additionally, IgG4-RD patients have plasma exosomes containing numerous differentially expressed proteins involved in complement activation and B cell differentiation[11]. However, the specific pathway that causes dysregulated B cells remains unclear.

The glycolysis/gluconeogenesis pathways are significantly upregulated in plasma B cells of IgG4-RD patients, suggesting abnormal metabolism[6]. Additionally, IgG4-RD causes mitochondrial dysfunction[12,13]. However, any abnormal metabolism as well as the potential mechanism in B cells of IgG4-RD is not fully understood.

To date, no standard diagnostic criteria exist for IgG4-RD. Diagnosis is through a comprehensive analysis of clinical, serological, radiological, and pathological data, which is tedious, time-consuming, and prone to misdiagnoses[14]. To address this, our study focused on discovering specific molecules for diagnosis that are involved in the pathogenesis of inflammation, fibrosis, immunodysregulation, and deregulated metabolism in B cells.

Using the bulk RNA-sequencing and single-cell RNA-sequencing, we identified ubiquitin-specific protease 25 (*USP25*) as a relevant gene that is down-regulated in IgG4-RD patients and involved in the pathogenesis. USP25 is a deubiquitinase belonging to the USP subgroup of the deubiquitinating enzyme (DUB) family that regulates the immune response and inflammation[15,16]. USP25 negatively regulates signaling induced by interleukin 17 (IL-17), an inflammatory mediator[17]. Research on the role of USP25 in fibrosis is still in its infancy. However, it is known that USP15 amplifies the fibrotic response by TGF-β signaling[18]. Interestingly, USP25 plays a crucial role in several metabolic diseases[19]. USP25 is a novel modulator of hypoxia-inducible factor-1α (HIF-1α) transcriptional activity and metabolic reprogramming, which suggests that USP25 is a chief regulator of glycolysis[20]. Considering this data and exploring the mechanistic role of USP25 in pathogenesis, we designed a study to determine the relationship between USP25 and IgG4-RD.

In this study, we demonstrate that the reduction of USP25 in IgG4-RD promotes IL-1β-mediated inflammation and SMAD3-induced fibrosis. In addition, the reduction of USP25 in IgG4-RD promotes RAC1 ubiquitination, which prevents ALDOA-mediated glycolysis. Together, our data suggest that USP25 is associated with fibrosis, inflammation, and metabolism of IgG4-RD patients, which serves as a potential diagnostic and therapeutic target.

## Results

### IgG4-RD patients have altered B cell homeostasis in the peripheral blood

Subpopulations of B cells in the peripheral blood mononuclear cells (PBMCs) of IgG4-RD patients were analyzed by flow cytometry to distinguish transitional B cells, plasma blast cells (PBC), memory B cells and naive B cells. Compared to healthy controls (HCs), IgG4-RD patients had increased percentages of PBC and memory B cells and decreased percentages of naive B cells and transitional B cells (Fig. 1A–C, S1A). Also in patients, there was a significant decrease in the mean fluorescence intensity (MFI) (Fig. 1D, S1B) and mRNA levels (Fig. 1E) of CD19 in all B-cell subsets except PBC. To verify the effect of CD19+ signal intensity on memory and naive B cells, we analyzed the MFI of IgD and CD27 in CD19+ cells. There were no significant differences in the MFI of IgD and CD27 in CD19+ cells in IgG4-RD patients compared to HCs (Fig. S1C, D). It is well known that the B cell activating factor receptor (BAFF-R) has an important function in B cell activation and survival[21,22]. The MFI of BAFF-R was found to be significantly reduced in PBMCs of patients compared to HCs (Fig.1F, S1E). Next, we found that there was no significant difference in the proportion of KI-67 (Fig. 1H, S1G) and apoptosis (Fig. 1G, S1F, S1H-I) of B cells. Furthermore, we examined the proliferation of PBMCs stimulated with CpG and biotin-F(ab′)$_2$ anti-human Ig (M + G). The results showed that the proliferation of B cells in patients was reduced compared to HCs (Fig. S1J). Finally, to exclude that these effects seen in IgG4-RD B cells were due to B cell receptor (BCR) density, we detected the intensity of CD79α and found no difference among HCs and patients (Fig. S1K, L). Collectively, these results suggest that the homeostasis of IgG4-RD B-cell subsets is disrupted, and the abnormal composition of B cells may have a key role in the pathogenesis of IgG4-RD patients.

### The early activation of B cells is reduced in IgG4-RD patients, while F-actin assembly is increased

To examine the effect of IgG4-RD on early B cell activation in naive (CD27-) and memory B cells (CD27 +), we used total internal reflection fluorescence microscopy (TIRFm) to examine BCR aggregation and B cell spreading following activation by membrane-tethered antigens (mAg) (Fig. S2A). Results showed that IgG4-RD naive B cells had reduced contact area at 5 min compared to IgG4-RD memory B cells and a reduced BCR MFI at 5 min compared to HC naive B cells (Fig. S2D–E). These results suggest that IgG4-RD impairs BCR aggregation and B cell spreading, both of which are critical for promoting BCR signaling. We next examined total BCR signaling via phosphotyrosine proteins (pY) and activation of Bruton's tyrosine kinase (pBTK), CD19 (pCD19) and SHIP (pSHIP). In IgG4-RD memory and naive B cells, the MFI of pCD19 (Fig. S2B&F), pY (Fig. 1I, K) and pBTK (Fig. 1J) was significantly decreased at 3 min compared to HC memory and naive B cells. At 5 min, IgG4-RD naive B cells had decreased MFI of pCD19 (Fig. S2F) and pY (Fig. 1K) and increased MFI of pSHIP (Fig. S2C&G) compared to HC naive B cells. In HCs, but not IgG4-RD, the MFI of pY (Fig. 1K) was significantly reduced in naive B cells at 3 min compared to memory B cells. In HCs and IgG4-RD, the MFI of pBTK (Fig. 1J) was significantly reduced at 3 min compared to their memory B cells. In addition, compared to HCs, the protein levels of pBTK were reduced, and the levels of pSHIP were increased in IgG4-RD PMBCs (Fig. 1L). Altogether, these results indicate that IgG4-RD inhibits the induction of strong positive BCR signaling in response to mAg stimulation.

Recent studies have demonstrated that the actin cytoskeleton participates in B cell spreading[23]. To test the effect of IgG4-RD on actin polymerization during mAg stimulation, we measured F-actin levels using phalloidin staining and also examined the recruitment of the actin nucleation factor, WASP, using anti-WASP antibody. We found that naive and memory B cells of IgG4-RD have enhanced MFI of F-actin (Fig. 1M, N) at 3 min and 5 min, and enhanced MFI of pWASP (Fig. 1O) at 3 min compared to HC naive and memory B cells. However, the MFI of F-actin in naive B cells of HCs and IgG4-RD patients at 5 min decreased compared to that of memory B cells (Fig. 1N). In addition, compared to HCs, the protein level of pWASP was elevated in PMBCs of IgG4-RD patients (Fig. 1P). Since the capacity of B cells to recognize and react to antigens is closely related to their morphology, we observed that the B cells of patients had decreased numbers (Fig. 1Q, R) and length (Fig. 1S) of filopodia compared to HCs. These results suggest that enhanced F-actin recruitment to the plasma membrane attenuates BCR clustering and signaling in IgG4-RD B cells.

### IgG4-RD patients have B cell activation deficiency corresponding to reduced USP25 expression

To investigate the transcriptomic changes in IgG4-RD patients, we performed transcriptome profiling of B cells isolated from HCs or IgG4-RD patients. KEGG enrichment networks indicated that the IL-17 signaling pathway may be critical in IgG4-RD pathogenesis (Fig. 2A, S2H–I). Furthermore, 611 (388 up-regulated, 223 down-regulated) significant DETs were identified between IgG4-RD and HC samples (Fig. S2J, Supplementary Data 1). Furthermore, it is noteworthy that a transcript of *USP25* exhibited significant down-regulation, and *USP25* emerged as a pivotal regulator within the IL-17 signaling pathway (Fig. S2K). It has been reported that the expression of IL-17 in the serum and tissues of IgG4-RD patients is increased and that USP25 negatively regulates the signaling and inflammation induced by IL-17[17,24,25]. These results imply that USP25 may be involved in the pathogenesis of IgG4-RD. To reinforce the bulk transcriptomics data, we performed the single-cell RNA-sequencing of PBMCs from HCs and IgG4-RD patients. A total of 46,528 cells were retained for subsequent analysis after filtering out doublets and poor-quality cells, including those that were dead or dying. Employing unsupervised clustering followed by a two-dimensional uniform manifold approximation and projection (UMAP) and utilizing known marker genes, we identified 6 major cell populations (Fig. S3A-D). For further in-depth analysis, we performed subclustering within the B cell population and 11 B cell subsets were identified[26] (Fig. 2B–D, S3E, F, S4A–G). To facilitate further analysis,

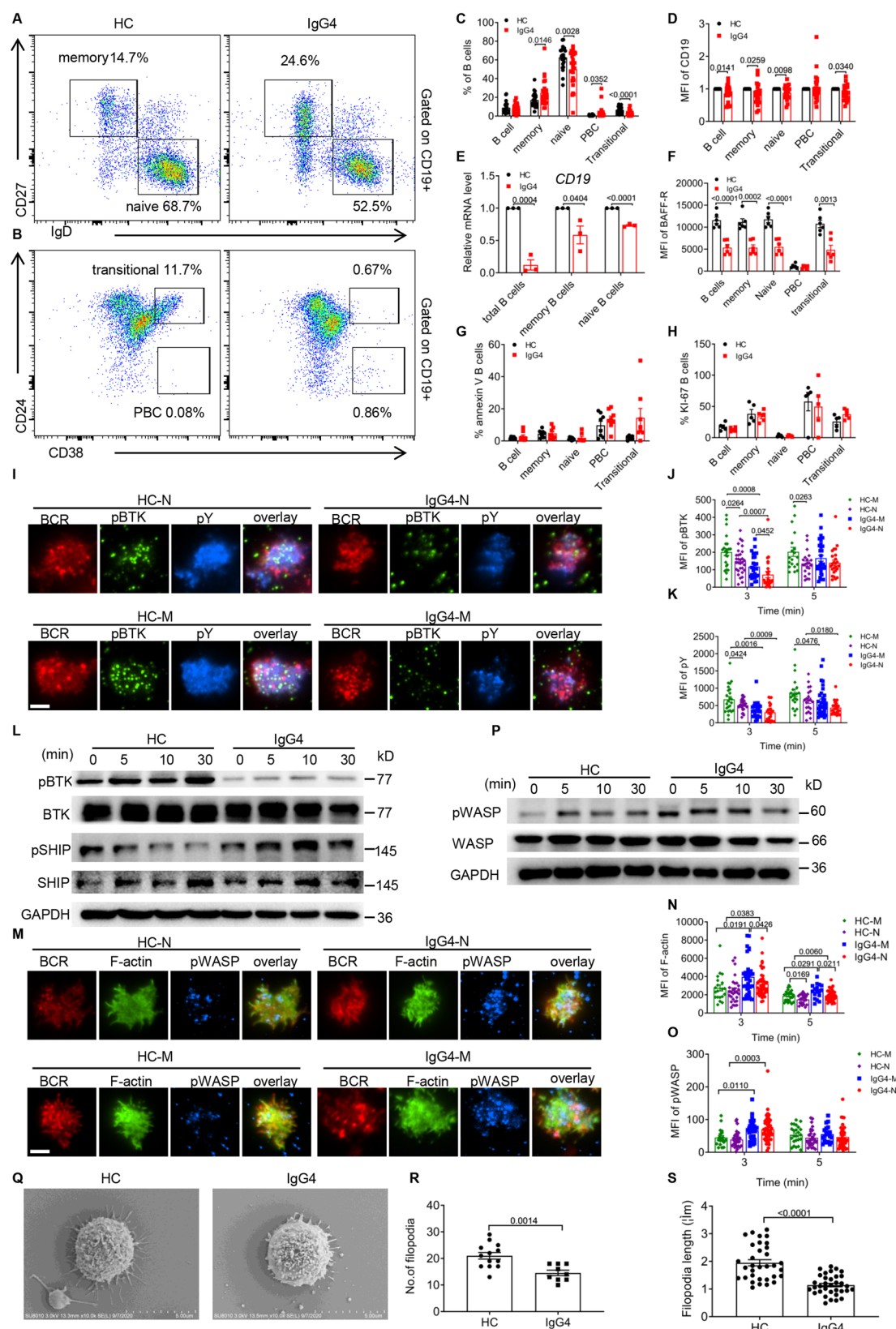

some B cell subsets with less than 20 cells would be combined with other B cell subsets. So, we combined transitional type 1 B cells (BT1), transitional type 2 B cells (BT2), and transitional type 3 B cells (BT3) into transitional B cells (BT) since BT1 and BT2 were few in IgG4-RD samples and BT3 were few in HCs samples. We also merged pre-switched memory B cells (BM-PS) into unswitched memory B cells (BM-

US) due to the low counts in IgG4-RD. After merging, we drew heat maps comparing naive B cell subsets between IgG4-RD and HCs, as well as memory B cell subsets and we found that naive B cells showed more expression changes of DEGs than other B cell subsets (Fig. S4A-G). In IgG4-RD, *USP25* expression was significantly decreased in resting naive B cells (BN-R). Activated naive B (BN-A) cells were nearly all in IgG4-RD,

**Fig. 1 | IgG4-RD patients have altered B cell homeostasis in the peripheral blood. A–D** Flow cytometry analysis of subpopulations of B cells from PBMCs of HCs and patients. The percentages of B cell subsets (**C**) (HCs=24, patients=25) and MFI of CD19 (**D**) (*n* = 21) were analyzed in PBMCs from HCs and patients. **E** Relative mRNA level of *CD19* was measured in total B cells, memory B cells and naive B cells from HCs and patients after sorting by RT-PCR (*n* = 3). **F–H** Analysis of the MFI of BAFF-R (**F**) (*n* = 6), the percentages of annexin V (**G**) (*n* = 8) and KI-67 (**H**) (*n* = 5) of the subpopulations of PBMCs from patients and HCs by Flow cytometry. **I–K** Memory B (CD27⁺) cells and naive (CD27⁻) B cells of HCs and patients were incubated with tethered to lipid bilayers for 3 and 5 min at 37 °C. Then cells were fixed, permeabilized and stained for AF647 anti-CD27, pBTK, and pY. The B cells were analyzed for the MFI of pY (**K**) and pBTK (**J**) by TIRFm (scale bar = 2.5 μm). **L** Western blot showing the levels of pBTK, pSHIP, BTK, and SHIP in B cells of patients and HCs. Representative results are shown. **M–O** B cells of patients and HCs were treated in the same way as in (**I**) except stained for pWASP and F-actin. The MFI of F-actin (**N**) and pWASP (**O**) were analyzed by TIRFm (scale bar = 2.5 μm). **P** Immunoblot showing the levels of pWASP and WASP in PBMCs of patients and HCs. Representative results are shown. **Q–S** Filopodia dilatation visualized using SEM (**Q**). Cells were analyzed for the number (**R**) (HCs=13, patients = 9) and length (**S**) (HCs = 33, patients = 34) of filopodia (scale bar = 5 μm). All images were representative images from 3 independent experiments. Data points in (**C–H, J, K, N, O, R**, and **S**) are represented as Mean ± SED. Statistical significance was based on two-tailed unpaired Student's t-test. Relevant p values are given in the graph. *$P < 0.05$; **$P < 0.01$; ***$P < 0.001$. Source data are provided as a Source Data file.

and the *USP25* expression level in them was lower than in resting naive B cells of HCs. In nearly all memory B cells, including BM-PS, resting switched (BM-SR), activated switched memory B cells (BM-SA), and age-associated B cells (ABC), *USP25* expression levels in IgG4-RD were observed to be lower than HCs, albeit without statistical significance. In BT1, BT2, and BM-US, *USP25* expression levels in IgG4-RD were higher than HCs without statistical significance, but these B cell subsets in IgG4-RD were diminished (Fig. 2E, S3F). Additionally, we found a significant reduction in *USP25* mRNA expression in total naive B cells of IgG4-RD patients (Fig. S4H-L). Furthermore, we validated these results by conducting RT-PCR, which confirmed that *USP25* mRNA expression was significantly reduced in both total memory and naive B cells after sorting (Fig. 2F). These findings indicate a predominance of B cells in both the naive and memory gates of IgG4-RD, with these disease-associated subsets exhibiting reduced expression of USP25.

To explore the role of USP25 deficiency in IgG4-RD patients, we used confocal microscopy to examine the spatial relationship between USP25 and BCR. We found that after stimulation of B cells with sAg, the colocalization between BCR and USP25 in B cells of IgG4-RD patients was significantly decreased at 5 min (Fig. 2G, H). Additionally, the protein levels of USP25 were significantly decreased in B cells of IgG4-RD patients compared to HCs (Fig. 2I, J). Previous studies have shown that BCR signaling transduction can be initiated through the recruitment of the signaling complex composed of LYN, SYK, and BLNK. In contrast, the degradation of LYN by ubiquitination negatively regulates BCR signaling[27–29]. Compared to HCs, the levels of USP25, LYN, pSYK, and pBLNK were decreased in PMBCs of IgG4-RD patients (Fig. 2J). Furthermore, both mRNA (Fig. S2L) and protein (Fig. 2J) levels of BLNK were decreased in IgG4-RD B cells. It has been reported that agammaglobulinemia patients have *BLNK* mutations that contribute to defects in mature B cell development[30–32]. Sequencing of genomic DNA showed a heterozygous synonymous mutation in the *BLNK* gene (c.171 T > C) of IgG4-RD patients (Fig. S2M). We speculate that *BLNK* synonymous mutations in IgG4-RD patients result in reduced production of *BLNK* mRNA transcripts, which may be due to synonymous mutation-mediated alterations in the secondary structure of *BLNK* mRNA. To further research the effects of reduced USP25 in IgG4-RD, *Usp25* KO mice were used to establish a mouse model, which was confirmed to lack expression of USP25 in B cells (Fig.2K). In *Usp25* KO B cells, the MFI of pBTK (Fig. S5A, B) and pY (Fig. S5C) was reduced, while the MFI of pSHIP was significantly increased (Fig. S5E, F). Also, the colocalization between pBTK and BCR was significantly decreased in *Usp25* KO B cells (Fig. S5D), while the colocalization between pSHIP and BCR was significantly increased (Fig. S5G). Furthermore, the protein expressions of LYN, pSYK, pBLNK, pY and pBTK were significantly decreased in *Usp25* KO B cells, while the level of pSHIP and pWASP was significantly increased (Fig. 2L, S5H). These results demonstrate that USP25 deletion decreases BCR signaling, which is consistent with that of IgG4-RD patients. Investigating how USP25 regulates upstream BCR signaling events, Co-immunoprecipitation (CO-IP) showed that USP25 associates with LYN in WT splenic B cells of WT mice (Fig. 2M). We next

examined the ubiquitination levels of LYN in *Usp25* KO B cells and found they had increased ubiquitination of LYN compared to WT (Fig. 2N), indicating that the interaction between LYN and USP25 reduces ubiquitination of LYN, which might prevent LYN degradation. To confirm that there is an increase of LYN ubiquitination in IgG4-RD that reduces BCR signaling, IgG4-RD PBMCs were treated with the proteasome inhibitor MG132. MG132 restored the degradation of USP25, LYN, pAKT, pFOXO1, and pS6 to the level of HCs B cells in IgG4-RD (Fig. 2O, S5I). Similarly, MG132 treatment in *Usp25* KO B cells restored the levels of LYN, pCD19, pSHIP (Fig. S5J), pWASP (Fig. S5K, L) and F-actin (Fig. S5M) to that of WT B cells. These results suggest that USP25 inhibits LYN ubiquitination, thereby protecting LYN from degradation enhancing BCR signaling, and reducing actin accumulation.

To study the effect of USP25 deficiency on humoral immune responses, mice were infected with lymphocytic choriomeningitis virus (LCMV) and immunized with NP-KLH, respectively. We found that immunization of *Usp25* KO mice resulted in a significant decrease in the percentage and number of PC cells, but also resulted in a significant increase in the percentage and number of plasma blast cells (PBC) (Fig. S6A–D). However, there was no significant difference in the percentage and number of PC and PBC after LCMV infection (Fig. S6I–K). Additionally, the frequency of germinal center B (GCB) cells was decreased in *Usp25* KO mice after immunization (Fig. S6E, F) and infection (Fig. S6L, M), while the number of GCB cells was not significantly different (Fig. S6G&N). Notably, *Usp25* KO mice have reduced antibody affinity maturation (Fig. S6H). Furthermore, compared to WT mice, *Usp25* KO mice showed a decrease in the frequency (Fig. S6O, P) and number (Fig. S6Q) of dark zone (DZ) B cells and an increase in the frequency of light zone (LZ) B cells following LCMV infection. Taken together, it was shown that USP25 is essential for triggering the humoral immune response in immunized and infected *Usp25* KO mice.

## USP25 deficiency causes an increase of IgG1+ antibody-secreting cells and IgG1 secretion and mediates the upregulation of SMAD3 to induce fibrosis

We evaluated whether the phenotype of IgG1 (a homologue of human IgG4[33]) in *Usp25* KO mice resembles that of IgG4-RD, we found that following stimulation of *Usp25* KO B cells with anti-CD40/IL-4, the percentage of IgG1+ (Fig. 3A, B) and IgE+ B cells was increased, while the percentage of IgG3 +, IgG2b +, and IgA+ B cells showed no significant differences (Fig. S7A–C). Additionally, in stimulated *Usp25* KO B cells, we observed an increase in the percentage of IgG1+ PC and PBC, which correlated with an increase in secreted IgG1 in culture supernatants (Fig. 3A–D, S7D, E). We found that the expression of GLTγ1, a transcript related to IgG1 class switch recombination[34], was significantly enhanced in *Usp25* KO B cells stimulated by LPS (Fig. 3E). Interestingly, we also observed that the percentage of IgG1+ PBC was increased in *Usp25* KO mice after LCMV infection (Fig. 3F, G), while the percentage of IgG1+ PC was not significantly different (Fig. 3H). Surprisingly, we found that the secretion of IgG1 was increased in the

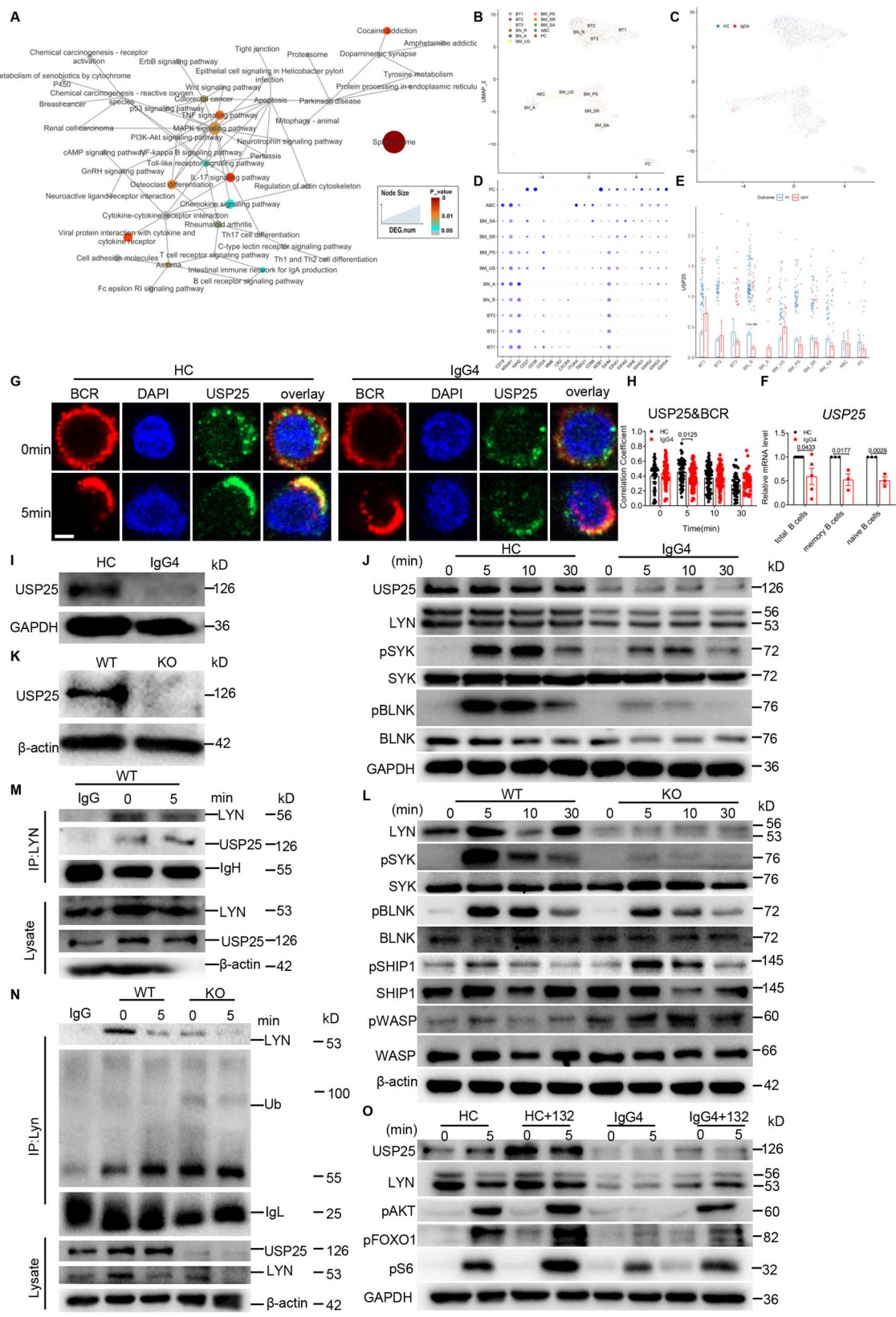

serum of both uninfected and infected *Usp*25 KO mice compared to WT mice (Fig. 3I). Moreover, the titer of NP-specific IgG1 was increased in *Usp25* KO mice compared to WT mice after immunization (Fig. 3J). To confirm the effect of reduced BCR signaling in *Usp25* KO on IgG1 expression, we conditionally deleted *Btk* in B cells by crossing *Btk*^fl/fl mice with *Mb1*^Cre mice (*Mb1*^Cre+/-*Btk*^fl/fl). Following anti-CD40/IL-

4 stimulation, we found significantly higher percentages of IgG1+ B cells in *Mb1*^Cre+/-*Btk*^fl/fl than in *Mb1*^Cre+/-*Btk*^fl/+ mice (Fig. 3K, L). We also observed that the percentage of IgG1+ PC and PBC was increased in *Mb1*^Cre+/-*Btk*^fl/fl stimulated B cells, which correlated with increased IgG1 secretion in culture supernatants (Fig. 3K, M, N, S7D&F). Not surprisingly, we found that LPS-stimulated B cells from *Mb1*^Cre+/-*Btk*^fl/f

**Fig. 2 | IgG4-RD patients have immunocompromise corresponding to reduced USP25 expression. A** Enriched KEGG pathway interaction networks in B cells from HCs and patients (*n* = 5). **B, C** Integration analysis results of IgG4-RD and HCs, showing the B cell clusters (**B**) and status (**C**) (*n* = 3). **D** Dot plot showing the expression of selected canonical cell markers in the 11 B cell subsets in patients and HCs. **E** The expression level of *USP25* on B cell subsets in IgG4-RD and HCs showed by UMAP separately (*n* = 3). **F** Relative mRNA level of *USP25* was measured in total B cells (*n* = 5), memory B cells (*n* = 3) and naive B cells (*n* = 3) from HCs and patients after sorting. **G, H** Purified B cells of patients and HCs were stimulated, fixed, permeabilized, and stained for USP25 (scale bar = 2.5 μm). The correlation coefficient between BCR and USP25 was quantified using more than 47–101 individual cells (**H**). **I** Western blots of USP25 in purified B cells from HCs and patients. **J** Western blots of USP25, LYN, pSYK, SYK, pBLNK and BLNK in PBMCs. **K** Western blots of USP25 levels in purified B cells from WT and *Usp25* KO mice. **L** Western blots of LYN, pSYK, SYK, pBLNK, BLNK, pSHIP, SHIP, pWASP, and WASP in B cells of WT and *Usp25* KO. **M** Immunoassay of lysates of WT cells treated with 10 μg/ml sAg for 5 min, followed by immunoprecipitation with LYN and immunoblot analysis of USP25. **N** Immunoassay of lysates of WT and *Usp25* KO B cells treated in the same way as in (**M**), followed by immunoprecipitation with LYN and immunoblot analysis with Ubiquitin. **O** Immunoblot showing the levels of USP25, LYN, pAKT, pFOXO1 and pS6. PBMCs were treated with 25 μM of MG132 at 37 °C for 30 min before incubating in the same way as in (Fig. 1J) for 5 min. All images were representative images from 3 independent experiments. Data points in (**E**, **F**, and **H**) are represented as Mean ± SED. Statistical significance was based on two-tailed unpaired Student's t-test and wilcoxon rank sum test. Relevant p values are given in the graph. *$P < 0.05$; **$P < 0.01$; ***$P < 0.001$. Source data are provided as a Source Data file.

mice had enhanced expression of GLTγ1 (Fig. 3O), but there was no significant difference in the levels of IgG1 in their serum, which is consistent with that of *Usp25* KO mice (Fig. 3P). Altogether, this suggests that reduced BCR signaling in *Usp25* KO mice affects IgG1 expression.

Next, we wanted to determine whether the downregulation of USP25 of IgG4-RD is involved in fibrosis. Using Masson staining, we found that *Usp25* KO mice had obvious fibrosis in the pancreas, kidney, and liver (Fig. 4A–D), but not in the lung (Fig. S7G, H). Interleukin-13 (IL-13), which is responsible for fibroblast activation[35], is upregulated in tissues of IgG4-RD patients. Similarly, we found a significant increase of IL-13 levels in the serum of *Usp25* KO mice (Fig. 4E). TGF-β1 has been reported to induce fibrosis through activation of SMAD3 signaling and it is known that *USP* inhibitors decrease TGF-β signaling in fibroblast cell lines[18]. Considering this, we tested whether USP25 reduction in IgG4-RD has a role in the pathogenesis of fibrosis by decreasing ubiquitin-mediated degradation of SMAD3. We examined the effect of MG132 on soluble antigen-stimulated IgG4-RD B cells using confocal microscopy. Untreated IgG4-RD B cells had increased MFI of pSMAD3, but after 5 min of MG132 treatment, the pSMAD3 MFI matched that of HCs B cells (Fig. 4F, G, S7I–K). Moreover, we found increased levels of pSMAD3, fibronectin (FN) and COL1A1 in IgG4-RD patients (Fig. 4H) and *Usp25* KO B cells (Fig. S7L). This indicates that down-regulation of USP25 induces SMAD3 up-regulation in B cells, leading to fibrosis in IgG4-RD patients.

### Inflammation in IgG4-RD patients is mediated via the IL-1β inflammasome axis

IgG4-RD has been reported as a systemic chronic inflammatory disease in which increased levels of most soluble receptors of the IL-1 family mediate inflammation[36]. We investigated the specific molecules involved in B cell induced inflammation in IgG4-RD patients. There was a significant elevation in the levels of NLRP3, caspase-1, IL-1β, pNF-κB and pIKKB in PBMCs of IgG4-RD patients (Fig. 4I, J) and *Usp25* KO (Fig. S8A), while the expression of TRAF6 was decreased. In examining the inflammatory phenotype of B cells from *Usp25* KO mice, we found elevated levels of IL-1β in their serum (Fig. 4K). In addition, after being injected intraperitoneally with LPS, *Usp25* KO mice had significant lymphocyte infiltration in the lungs (Fig. 4L), but there was no significant difference in the colon, liver, or kidney (Fig. S8B). *Usp25* KO mice also had significantly elevated levels of caspase-1, NLPR3, and IL-1β following LPS injection (Fig. 4M). Stimulation with LPS, CpG or soluble antigen (sAg) revealed a reduction in the proliferation of *Usp25* KO B cells stimulated with CpG or sAg compared to WT B cells (Fig. S8C), and the apoptosis of B cells had no significant difference from WT mice (Fig. S8D, E). Additionally, the percentage of KI-67 + B cell subsets was significantly lower in *Usp25* KO mice (Fig. S8F–H), while the percentage of annexin-V + B cell subsets had no significant difference (Fig. S8I, J). Finally, CD19 immunohistochemical staining showed that more B cells were present in the pancreas, kidney, liver and lung of

*Usp25* KO mice (Fig. S8K). These results indicate that down-regulated USP25 in B cells of IgG4-RD patients induces inflammation through the IL-1β inflammatory axis.

### USP25 deficiency affects B cell metabolism in IgG4-RD patients through the PI3K-AKT pathway

To further explore the potential mechanism concerning the participation of B cell metabolism in the pathogenesis of IgG4-RD, we examined the changes in PI3K/AKT-mediated signaling pathways. We found that PI3K/AKT-mediated downstream metabolic signaling molecules including pPI3K, pAKT, pmTOR, pS6, pFOXO1 had reduced activation in PBMCs of IgG4-RD patients compared to HCs (Fig. 5A, S9A), which was also consistent with the results in *Usp25* KO B cells (Fig. S9B). Moreover, the expression of the glycolytic isoenzyme, PKM2, was also reduced (Fig. 5A). It has been shown that c-MYC and HIF-1α regulate gene expression of the enzymes involved in the TCA cycle and glycolysis, while the transcription factor STAT5 affects cellular metabolism by regulating HIF-1α[37–39]. We detected that the protein levels of c-MYC, HIF-1α, and pSTAT5 were all reduced in IgG4-RD patients (Fig. 5A) and *Usp25* KO B cells (Fig. S9B). Additionally, we found that the basal and maximal respiration rates of B cells from IgG4-RD patients were reduced compared to HCs B cells (Fig. 5B), while *Usp25* KO B cells had only reduced maximal respiration rate (Fig. S9C), but both IgG4-RD patients B cells and *Usp25* KO B cells had significantly reduced glycolytic capacity (Fig. 5C, S9D). Next, transmission electron microscopy (TEM) revealed that the mitochondria were significantly enlarged in IgG4-RD PBMCs (Fig. 5D) and *Usp25* KO B cells (Fig. S9E). Together, these results indicate that in IgG4-RD patients reduced USP25 expression leads to mitochondrial dysfunction in B cells. To further investigate the effect of IgG4-RD on B cell metabolism, mitochondrial morphological changes were observed by staining the ER and mitochondria and analyzing them by confocal microscopy. Compared to HCs B cells, IgG4-RD B cells showed significantly reduced MFI of Mito Tracker and ER (Fig. 5E–G), as well as significantly reduced colocalization of ER/BCR and Mito/BCR after sAg activation (Fig. 5H–I). We also observed the mitochondria morphology in B cells by confocal microscopy. The staining of PK Mito was significantly reduced in IgG4-RD patients (Fig. 5J, K) and *Usp25* KO (Fig. 5L, S9F) B cells compared to HCs and WT B cells, respectively. Additionally, by flow cytometry, we found that mitochondrial membrane potential changes (Fig. 5M, S9G–I) and the intracellular levels of reactive oxygen species (ROS) (Fig. 5N, S9J, K) were lower in IgG4-RD and *Usp25* KO B cells compared to HCs and WT B cells, respectively. The metabolomic profiles exhibited significant changes between HCs and IgG4-RD patients[40]. To further investigate the difference in serum metabolic composition between *Usp25* KO and WT mice, we performed a metabolic analysis. By the PLS-DA model, 455 differentially expressed metabolites, including 210 downregulated and 245 upregulated, were altered in the serum of *Usp25* KO mice. Pathways were enriched and visualized for differentially expressed metabolites involving the multiple metabolic

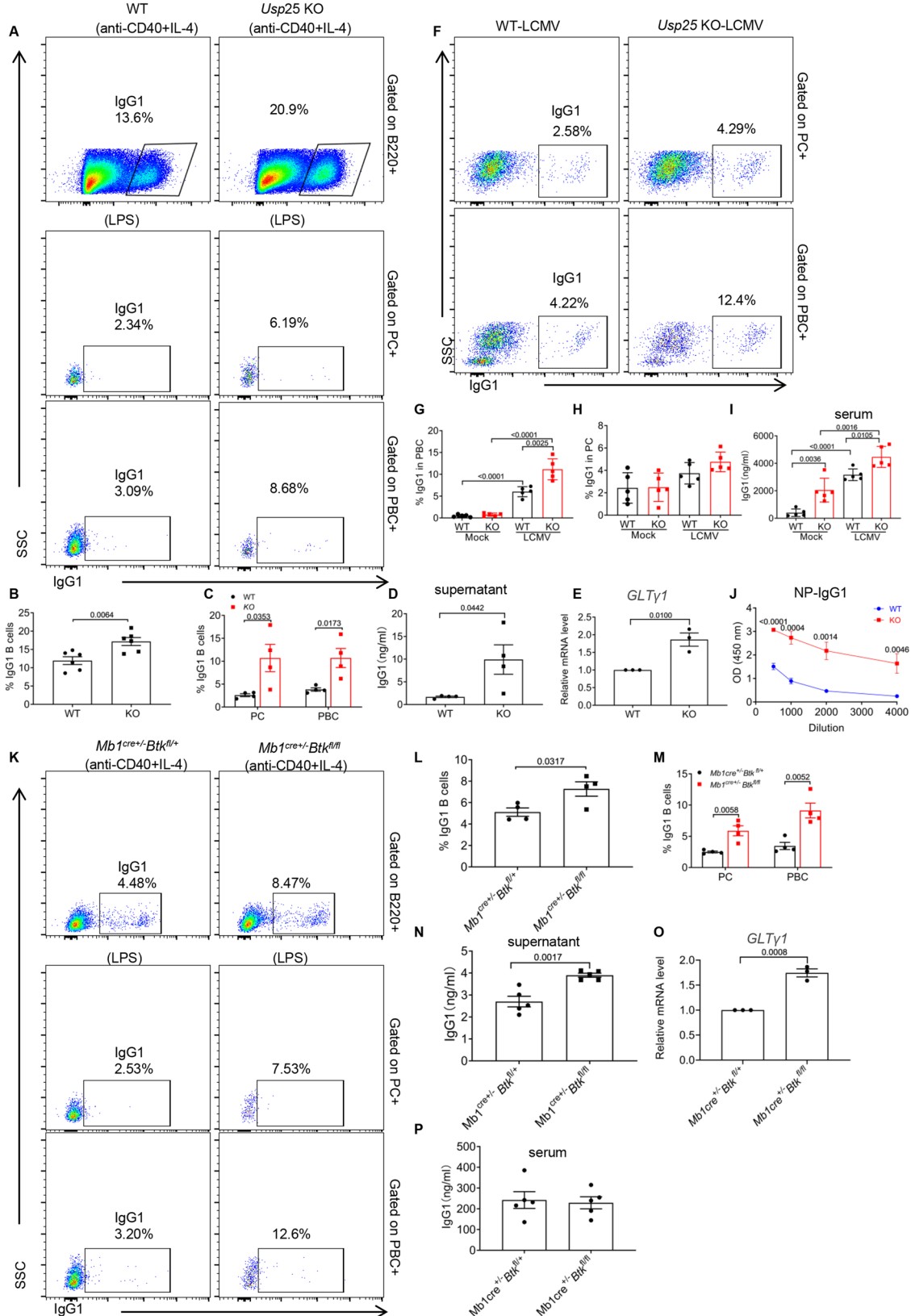

pathways: alpha-linolenic and linoleic acid metabolism, which were also enriched in IgG4-RD[40] (Fig. 5O, P). Moreover, we measured the increase of glutamic and the decrease of cysteine in IgG4-RD and *Usp25* KO serum (Fig. 5Q–T). In conclusion, these results suggest that in IgG4-RD patients the USP25 deficiency affects B cell metabolism by inducing mitochondrial dysfunction through the PI3K-AKT axis.

## USP25 interacts with RAC1 to regulate the glucose metabolism in B cells

The *Usp25* KO mice only mimic the IgG4-RD patients' clinical symptoms to a certain extent. Therefore, transcriptome analysis of B cells isolated from mice was performed to further explore the differences in the transcriptomic characteristics of the *Usp25* KO mice model. The

**Fig. 3 | USP25 deficiency causes the increase of IgG1. A–C** Flow cytometry analysis of IgG1 in B cells, PC and PBC from WT and *Usp25* KO mice after stimulating with 8 ng/ml IL4 plus 10 μg/ml of CD40 for 120 h and 10 μg/ml of LPS for 72 h and shown are representative dot plots (**A**). The percentages of B cells (**B**), PC and PBC (**C**) were analyzed from WT and *Usp25* KO mice (*n* = 6). **D** The level of IgG1 in PC and PBCs culture supernatant of WT and *Usp25* KO mice was quantified by ELISA (*n* = 4). **E** B cells of *Usp25* KO and WT mice were cultured with LPS (10 μg /mL). Total RNA was isolated after 2 days of culture and the GLTγ1 and GAPDH levels were measured by RT-PCR (*n* = 3). **F-H** Flow cytometry analysis of IgG1 in PBC (**G**) and PC (**H**) from *Usp25* KO and WT mice after LCMV infection (*n* = 5). Shown are representative dot plots (**F**). **I** Quantification of IgG1 antibody levels in the serum of WT and *Usp25* KO mice after infection with LCMV by ELISA (*n* = 5). **J** The OD of NP-IgG1 after

immunized mice was quantified using ELISA (*n* = 4). **K–M** Flow cytometry analysis of IgG1 in B cells (**L**), PC and PBC (**M**) from *Mb1^cre+/-^Btk^fl/+^* and *Mb1^cre+/-^Btk^fl/fl^* mice after stimulating in the same way as in (3 A) (*n* = 4). Shown are representative dot plots (**K**). **N** The level of IgG1 in PC and PBC culture supernatant of *Mb1^cre+/-^Btk^fl/+^* and *Mb1^cre+/-^Btk^fl/fl^* mice was quantified by ELISA (*n* = 5). **O** The GLTγ1 level was measured in the same way as in (3E) in *Mb1^cre+/-^Btk^fl/+^* and *Mb1^cre+/-^Btk^fl/fl^* mice (*n* = 3). **P** IgG1 antibody levels in the serum of *Mb1^cre+/-^Btk^fl/+^* and *Mb1^cre+/-^Btk^fl/fl^* mice were quantified by ELISA (*n* = 5). Data points in (**B, C, D, E, G, I, J, L, M, N,** and **O**) are represented as Mean ± SED. Statistical significance was based on two-tailed unpaired Student's t-test. Relevant p values are given in the graph. *$P < 0.05$; **$P < 0.01$; ***$P < 0.001$. Source data are provided as a Source Data file.

KEGG-enriched pathways and DETs heat map showed that *Rac1*-related pathways (asterisk mark) were markedly enriched and two *Rac1* transcripts were simultaneously significantly down-regulated in *Usp25* KO mice (Fig. 6A, B). Intriguingly, IgG4-RD patients present concordant *Rac1* down-regulation and *Rac1*-related pathways enrichment with those in *Usp25* KO mice (Fig. 6C, S10A). Moreover, the mRNA and protein levels of RAC1 were both reduced in B cells from IgG4-RD patients and *Usp25* KO mice (Fig. 6D, E). It has been reported that *Rac1* is targeted by the ubiquitin-proteasome pathway to regulate cell polarity and cytoskeletal dynamics[41,42]. To verify the specific mechanism of RAC1 reduction in IgG4-RD, we first used CO-IP to verify that USP25 and RAC1 interact in WT B cells (Fig. 6F). We next examined the ubiquitination levels of RAC1 in IgG4-RD patients and found they had increased ubiquitination of RAC1 compared to HCs (Fig. 6G). Corresponding to this, there were significantly decreased levels of RAC1 in IgG4-RD patients (Fig. 6H–I) and the colocalization between RAC1 and LAMP1 (late lysosomal marker) was significantly increased after sAg stimulation (Fig. 6J). When treated with lysosomal activity inhibition, Bafilomycin A1, the RAC1 level of IgG4-RD patients was restored to some extent to that of HCs (Fig. S10B, C). Altogether, these results suggest that interaction between RAC1 and USP25 can reduce the ubiquitination of RAC1 and prevent its transport to lysosomes for degradation.

It has been reported that activated RAC1 binds PI3K, which recruits VAV2 to activate RAC1. This PI3K signaling-mediated activation of RAC1 induces the breakdown of F-actin that then releases and activates aldolase A (ALDOA) to promote up-regulation of the glycolytic pathway[43–45]. Next, we observed an increase in F-actin levels and F-actin/ALDOA colocalization in *Usp25* KO B cells (Fig. 6K–M), which is also the same result as in TIRFm (Fig. 7A–E). To measure the soluble versus the immobilized state of ALDOA, we permeabilized B cells with digitonin to allow diffusive ALDOA efflux and then separated the collection of supernatant and cell lysates for ALDOA analysis. The results showed that the levels of ALDOA in the supernatant of PBMCs from IgG4-RD patients were reduced after stimulation with sAg, while there was no significant difference in cell lysates (Fig. 7F). Comparable results were observed in *Usp25* KO B cells as well (Fig. 7G). Next, we inhibited actin polymerization with latrunculin B, which didn't affect cellular glucose uptake. Intriguingly, we found that F-actin (Fig. 7H, S10D) and pWASP (Fig. 7I, S10E) levels in IgG4-RD B cells could be restored to that of HCs. The same was seen with ROS when actin polymerization was inhibited (Fig. 7J, S10F). These results confirm that RAC1 can release F-actin-bound ALDOA, increasing ALDOA activity to regulate glycolysis. To further explore the effect of soluble ALDOA on glycolysis, we detected that the expression of downstream *ALDOA* genes, including *TP1, ECON,* and *PKM2*, were significantly reduced during glycolysis in IgG4-RD patients (Fig. 7K), which was also seen in *Usp25* KO mice (Fig. S10G). These results indicate that in IgG4-RD B cells, the interaction of USP25 with RAC1 is decreased, resulting in increased ubiquitination of RAC1, which reduces ALDOA release from the cytoskeleton and thus, also reduces glycolysis.

## Overexpressing USP25 in IgG4-RD cell lines rescues the clinical phenotype of IgG4-RD patients

To confirm the role of USP25 in regulating the clinical phenotype of IgG4-RD patients, we constructed an IgG4-RD cell line overexpressing USP25. This cell line was made from Epstein-Barr virus (EBV) induced conversion of HC and IgG4-RD human peripheral blood B lymphocytes into lymphoblastoid cells. Intriguingly, USP25 levels were significantly higher in the cell lines compared to HCs and IgG4-RD cells and significantly higher in the IgG4-RD cell line compared to the HC cell line (Fig. 7L). Moreover, compared to the HC cell line, the ROS level of the IgG4-RD cell line was increased after stimulation with sAg (Fig. 7M). And results of ECAR showed that the glycolytic ability of the IgG4-RD cell line was restored to the level of the HC cell line (Fig. 7N). Next, we found increased levels of USP25 in the IgG4-RD cell line by western blot and phosflow (Fig. 7O, S10H), along with increased levels of RAC1 (Fig. S10I), but decreased levels of pWASP (Fig. S10J), F-actin (Fig. S10K) and FN (Fig. S10L). Additionally, BCR signaling molecules pSYK and pBLNK, and metabolism-related molecules pS6, pmTOR, and c-MYC were increased in levels, while fibrosis-related protein COL1A1 and inflammatory signaling molecule NLRP3 were decreased (Fig. 7O). Next, we inhibited the expression of USP25 with the USP25 inhibitor AZ1. Interestingly, we found that the expression of USP25, LYN, pSYK, pmTOR, pS6, pFOXO1, c-MYC, COL1A1, and RAC1 could be restored to the level of the HC cell line (Fig. 7P). Furthermore, we overexpressed USP25 in IgG4-RD cell line, which has reduced USP25 expression, by lentiviral transfection of USP25 or the empty vector, followed by western blot to detect that the protein levels of USP25, pBTK, LYN, pSYK, RAC1, PKM2, and pSMAD3 were all rescued to the levels of HC cell line by western blot (Fig. 7Q).

In conclusion, these results all suggest that USP25 is involved in regulating BCR signaling, metabolism, fibrosis, and inflammation in IgG4-RD.

## Discussion

In this study, we have systematically analyzed the underlying molecular mechanism of the main clinical phenotypes of IgG4-RD patients. We have identified that USP25 is involved in regulating the main clinical phenotypes and that the *Usp25* KO mouse model is useful for mimicking IgG4-RD. When analyzing the BCR signaling of IgG4-RD patient B cells, we found that overall BCR signaling is decreased. This corresponds with the finding that LYN, a key upstream BCR signaling molecule, has reduced expression, which is caused by ubiquitin-mediated degradation. The bulk RNA-sequencing data uncovered the involvement of the IL-17 signaling pathway in the observed inflammation in IgG4-RD patients. Significantly, our analysis identified USP25 as a key player in this pathway. Transcriptomic differences thus defined reflect expansions of activated B cell subsets and these types of trancriptomic changes will be seen not just in many different inflammatory diseases (beyong IgG4-RD) but even after immunization and infection. Thus, it may be accurate to sort the major B cell subpopulations and then perform transcriptomic analyses. To validate the

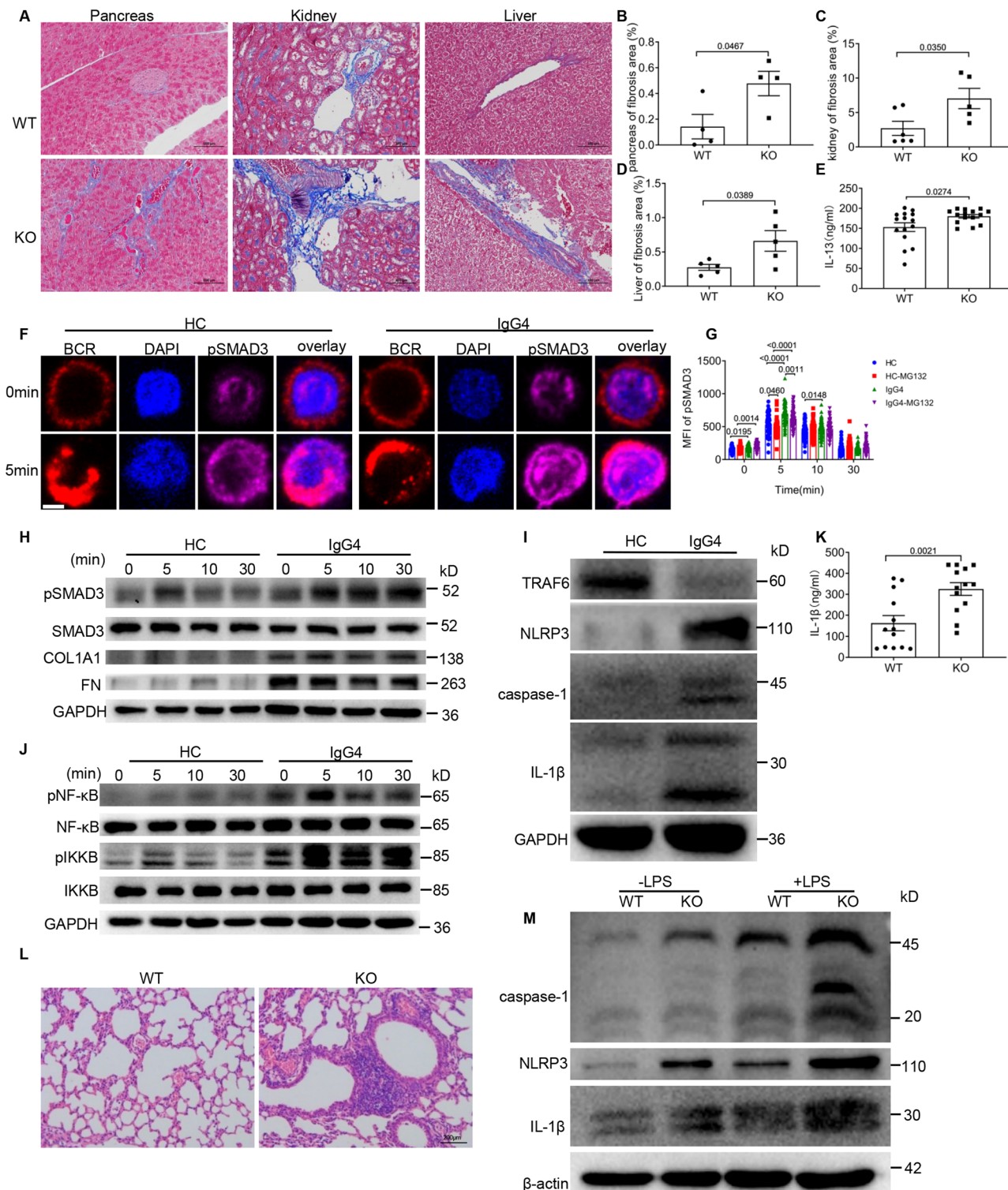

**Fig. 4 | Inflammation in IgG4-RD patients is mediated via the IL-1β inflammasome axis. A–D** Masson staining of pancreatic (**B**, $n = 4$), kidney (**C**, $n = 5$), and liver (**D**, $n = 5$) tissues and the percentage of the fibrotic area from WT and *Usp25* KO mice (scale bar = 100 μm). Shown are representative dot plots (**A**). **E** IL-13 levels in the serum of *Usp25* KO and WT mice were quantified by ELISA ($n = 15$). **F, G** Purified B cells from HCs and IgG4-RD patients were pre-incubated with 25 μM of MG132 at 37 °C for 30 min and then stimulated in the same way as in (Fig. 2G) except stained for pSMAD3 and nuclei using DAPI (scale bar = 2.5 μm) and shown are representative images captured using confocal (**F**). The MFI of pSMAD3 was analyzed (**G**). **H** Western blots of pSMAD3, SMAD3, COL1A1, and fibronectin (FN) in PBMCs of IgG4-RD patients and HCs stimulated with sAg for designated times. **I** Immunoblot showing the levels of TRAF6, caspase-1, NLRP3, and IL-1β in PBMCs

from HCs and IgG4-RD patients. **J** Western blots of pNF-κB, pIKKB, NF-κB, and IKKB in PBMCs from HCs and IgG4-RD patients stimulated with sAg for designated times. **K** IL-1β levels in the serum of *Usp25* KO and WT mice were quantified by ELISA ($n = 13$). **L, M** H&E staining results of lung tissues from WT and *Usp25 KO* mice (**L**) and western blot analysis of the levels of caspase-1, NLRP3, and IL-1β in B cells from WT and *Usp25* KO mice (**M**) after being injected (i.p.) with 10 mg/kg of LPS for 24 h (Scale bar, 200 μm). All images were representative images from 3 independent experiments. Data points in (**B**, **C**, **D**, **E**, **G**, and **K**) are represented as Mean ± SED. Statistical significance was based on two-tailed unpaired Student's t-test. Relevant p values are given in the graph. *$P < 0.05$; **$P < 0.01$; ***$P < 0.001$. Source data are provided as a Source Data file.

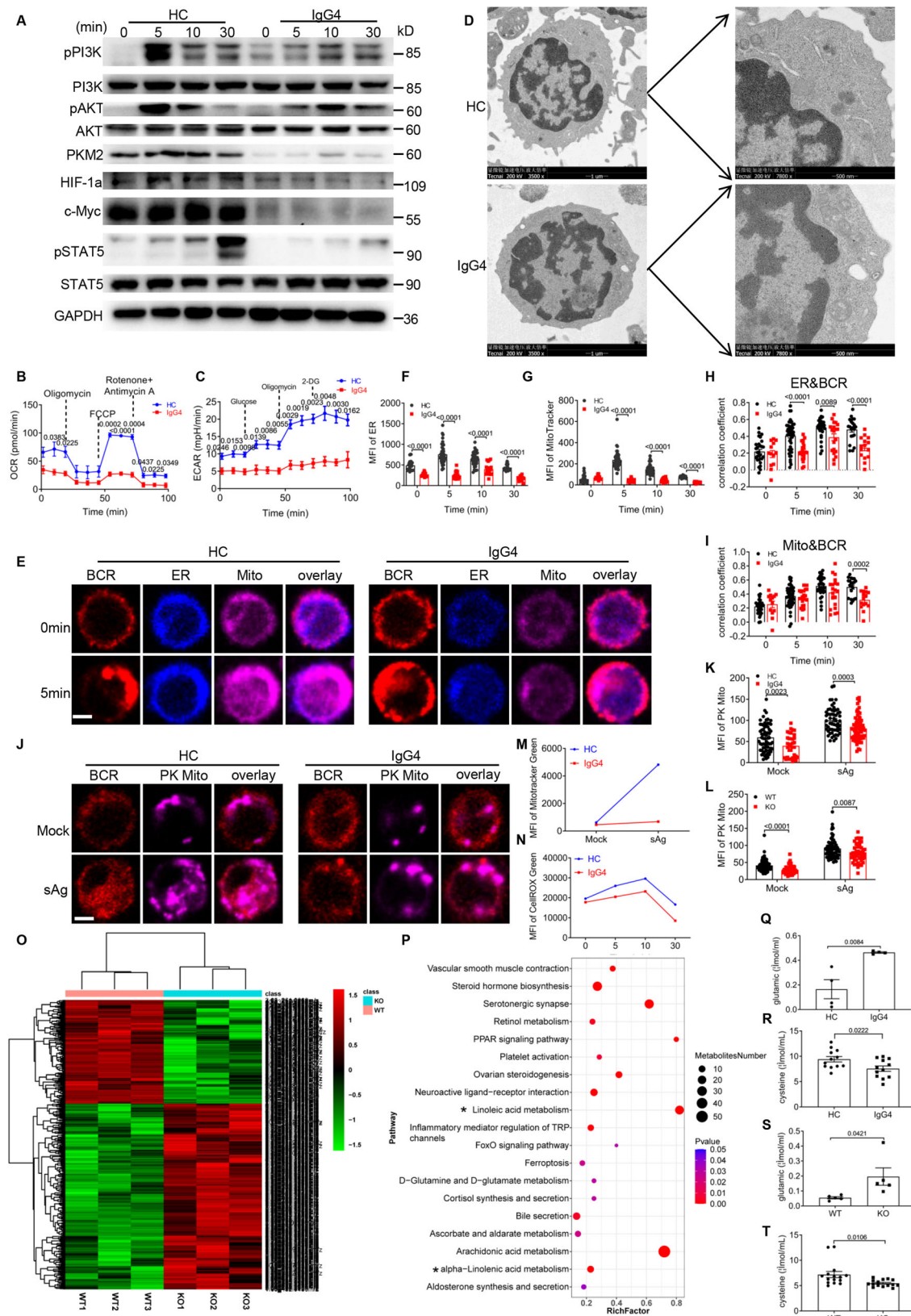

insights derived from bulk transcriptomics data, we conducted single-cell RNA-sequencing analysis on PBMCs collected from both HCs and IgG4-RD patients. It's noteworthy that the percentage of B cell sub-populations within naive and memory B cells showed no significant changes, which is different from our experimental observations (Fig. S3F). These discrepancies may be attributed to variations in sensitivity

between the two technologies, potential batch effects, sample size, and biological heterogeneity, among other contributing factors. However, *USP25* mRNA exhibited a notable reduction in both naive and memory B cells from IgG4-RD patients, aligning with our experimental findings. The expression of USP25 is also reduced in protein levels in IgG4-RD patients. We also identified that LYN is a substrate of USP25,

**Fig. 5 | USP25 deficiency affects B cell metabolism in IgG4-RD patients through the PI3K-AKT pathway. A** Western blot of pPI3K, pAKT, PI3K, AKT, PKM2, HIF-1α, c-Myc, pSTAT5 and STAT5 in PBMCs. **B, C** OCR (**B**) and ECAR (**C**) detection of B cells ($n = 3$). **D** The mitochondrial structure of B cells were analyzed by TEM (scale bar = 1 μm-500 nm). **E–I** Purified B cells were stimulated in the same way as in (Fig. 2G) except stained for Mito and ER (scale bar = 2.5 μm). The B cells were analyzed for MFI of ER (**F**) and Mito (**G**). The correlation coefficient of ER/BCR (**H**) and Mito/BCR was analyzed (**I**). **J, K** Purified B were pre-incubated with 10 μg/ml sAg for 24 h at 37 °C and then stained for anti-CD19 and PK Mito. Confocal analysis of the MFI of PK Mito in CD19+ cells (scale bar = 2.5 μm) (**K**). **L** Purified splenic B cells were stimulated with 10 μg/ml sAg for 24 h at 37 °C, stained for anti-B220 and mitotracker, and then analyzed for the MFI of mitotracker Green in B220 + B cells ($n = 3$). **M** PBMCs were stimulated in the same way as in (5 J) except stained for anti-CD19 and mitotracker, and then CD19 + B cells were analyzed for the MFI of mitotracker Green ($n = 3$). **N** PBMCs were stimulated, and stained for anti-CD19 and CellIROX Green, and then CD19 + B cells were analyzed for the MFI of CellIROX Green using flow cytometry ($n = 3$). **O** Cluster analysis of the differentially expressed genes in the serum of WT and *Usp25* KO mice is presented in the form of a heatmap ($n = 3$). **P** Top enriched up-regulated and down-regulated KEGG pathways for differentially expressed metabolite in the serum of *Usp25* KO and WT mice. **Q–T** The serum levels of glutamic ($n = 4$) and cysteine ($n = 15$) were detected in mice and patients respectively. All images were representative images from 3 independent experiments. Data points in (**B, C, F, G, H, I, K, L, Q, R, S**, and **T**) are represented as Mean ± SED. Statistical significance was based on two-tailed unpaired Student's t-test. Relevant p values are given in the graph. *$P < 0.05$; **$P < 0.01$; ***$P < 0.001$. Source data are provided as a Source Data file.

which is also reduced in IgG4-RD patients. We have established that the USP25-LYN axis regulates BCR signaling and that its underlying molecular mechanism accounts for the B cell activation deficiency. Decreased expression of LYN and the consequently reduced BCR signaling have also been confirmed in *Usp25* KO B cells. Although different antigenic stimulations with the help of different cells lead to different proportions of activated B cells differentiating into PBCs, USP25 is essential for triggering the humoral immune response in NP-KLH immunization and LCMV infection. Moreover, we have found a heterozygous synonymous mutation in *BLNK* of IgG4-RD patients, which is a critical adaptor for BCR signaling. The mutation may cause a reduction of both protein and mRNA levels of BLNK and consequently BCR signaling. Therefore, this point mutation of *BLNK* makes a considerable biomarker for precisely diagnosing IgG4-RD.

IgG4-RD patients have fibrosis in multiple organs. Also, *Usp25* KO mice have fibrosis in multiple organs. Studying this mouse model, we found that the key signaling pathway is via pSMAD3. This pathway is responsible for the secretion of FN and COL1A1, both of which are increased in B cells of *Usp25* KO mice and IgG4-RD patients. These results suggest that USP25 not only regulates BCR signaling but also controls fibrosis. Additionally, USP25 has been reported to be an inhibitor of inflammation[16,17]. Increased inflammation of IgG4-RD patients can be explained by the reduced expression of USP25 as well as the downstream molecule, TRAF6. Furthermore, we have examined the classical inflammation pathway and found that the signaling molecules NLRP3, caspase1, and IL-1β are all enhanced in IgG4-RD patients and *Usp25* KO mice. Therefore, we have established that USP25 is involved in regulating fibrosis and inflammation in IgG4-RD patients.

The last important clinical manifestation we looked at is the dysregulated metabolism in IgG4-RD. We have observed reduced Mito and ROS in both IgG4-RD and *Usp25* KO B cells. OCR and ECAR were decreased in both IgG4-RD and *Usp25* KO B cells. Furthermore, we also found that the mitochondria structure is distorted in both IgG4-RD and *Usp25* KO B cells. In addition, we found that the PI3K mediated mTORC1 and mTORC2 signaling pathways are reduced, and also key molecules in controlling metabolisms such as c-MYC, HIF1-α, and PKM2 are reduced. Mechanistically, we have found that the substrate of USP25, RAC1, regulates glycolysis and metabolism by depolymerizing actin and activating ALDOA, which is upstream of HIF1-α. Thus, we have unveiled a signaling pathway where USP25 interacts with RAC1 to regulate metabolism via regulating actin reorganization.

To further confirm that USP25 is involved in regulating the clinical phenotype of IgG4-RD patients, we constructed an IgG4-RD cell line that over-expresses USP25. Using this cell line, we found that BCR signaling was enhanced, metabolic molecules were activated, and fibrosis and inflammation molecules were reduced. Furthermore, we also constructed an IgG4-RD cell line that has reduced USP25 expression. Using this cell line to overexpress USP25 by lentiviral transfection of the USP25 plasmid, we found that BCR signaling, fibrosis, and activation of metabolic molecules are rescued. In summary, our study has

provided for the mechanism of pathogenesis in the clinical phenotype of IgG4-RD patients and we also propose possible target molecules to treat IgG4-RD patients.

So far, several mouse models have been established that have symptoms similar to IgG4-RD[46,47]. However, these mice have very limited lesion tissue, and the pattern of immune-related responses is not fully defined. The role of B cells in the pathogenesis of these mice is not fully understood. Therefore, we propose the *Usp25* KO mouse as an IgG4-RD model. In *Usp25* KO mice, increased IgG1 and IgE were detected in splenic B cells, and these mice have marked fibrosis in the pancreas, kidney, and liver. Additionally, we detected increased expression of several inflammatory factors in the serum of *Usp25* KO mice and found significant lymphocytic infiltration in the lungs after LPS stimulation in vivo. This study demonstrates inflammatory similarities between *Usp25* KO mice and IgG4-RD patients. These findings support the usefulness of *Usp25* KO mice as an experimental model for human IgG4-RD specifically for understanding the molecular mechanisms involved in the phenotype of IgG4-RD.

IgG4-RD is a distinct systemic autoimmune-mediated disease. So far, numerous studies have provided evidence that epigenetic mechanisms are found to be related to the risk of various autoimmune diseases. DNA methylation is a well-studied epigenetic modification involved in gene expression regulation[48]. Wu X, et al. found aberrant DNA methylation modifications in peripheral B cells of IgG4-RD patients by genome-wide DNA methylation, proving the potential role of DNA methylation changes in the pathogenesis of IgG4-RD[49]. Meanwhile, we found DNA methyltransferase-related or demethylation-related genes differentially expressed in peripheral B cells of IgG4-RD patients by the bulk RNA-sequencing, including MTRR, METTL23, TET2, USP9X, HEMK1. We speculate that the reduction of USP25 in IgG4-RD patients may be a result of changes in DNA methylation.

Overall, our study has identified USP25 is involved in immuno-dysregulation, fibrosis, inflammation, and metabolism of IgG4-RD. The reduction of USP25 in IgG4-RD patients is observed to cause the decreased expression of LYN and BCR signaling. Additionally, the decreased USP25 promotes the IL-1β-mediated classical inflammasome pathway and SMAD3-induced fibrosis. Intriguingly, the reduction of USP25 decreases RAC1 expression and ALDOA activation by enhancing actin polymerization, which prevents HIF1-α-mediated metabolism. Taken together, our research introduces the underlying molecular mechanism in B cells that contributes to the pathogenesis of the classical clinical symptoms of IgG4-RD patients and offers an optimal mouse model. These discoveries provide biomarkers as targets and animal models for the diagnosis and treatment of IgG4-RD.

## Methods

The research methods applied in this study followed the guidelines of the World Medical Association's Declaration of Helsinki and subsequent revisions, and the study was reviewed and approved by the Ethics Committee of the Huazhong University of Science and Technology.

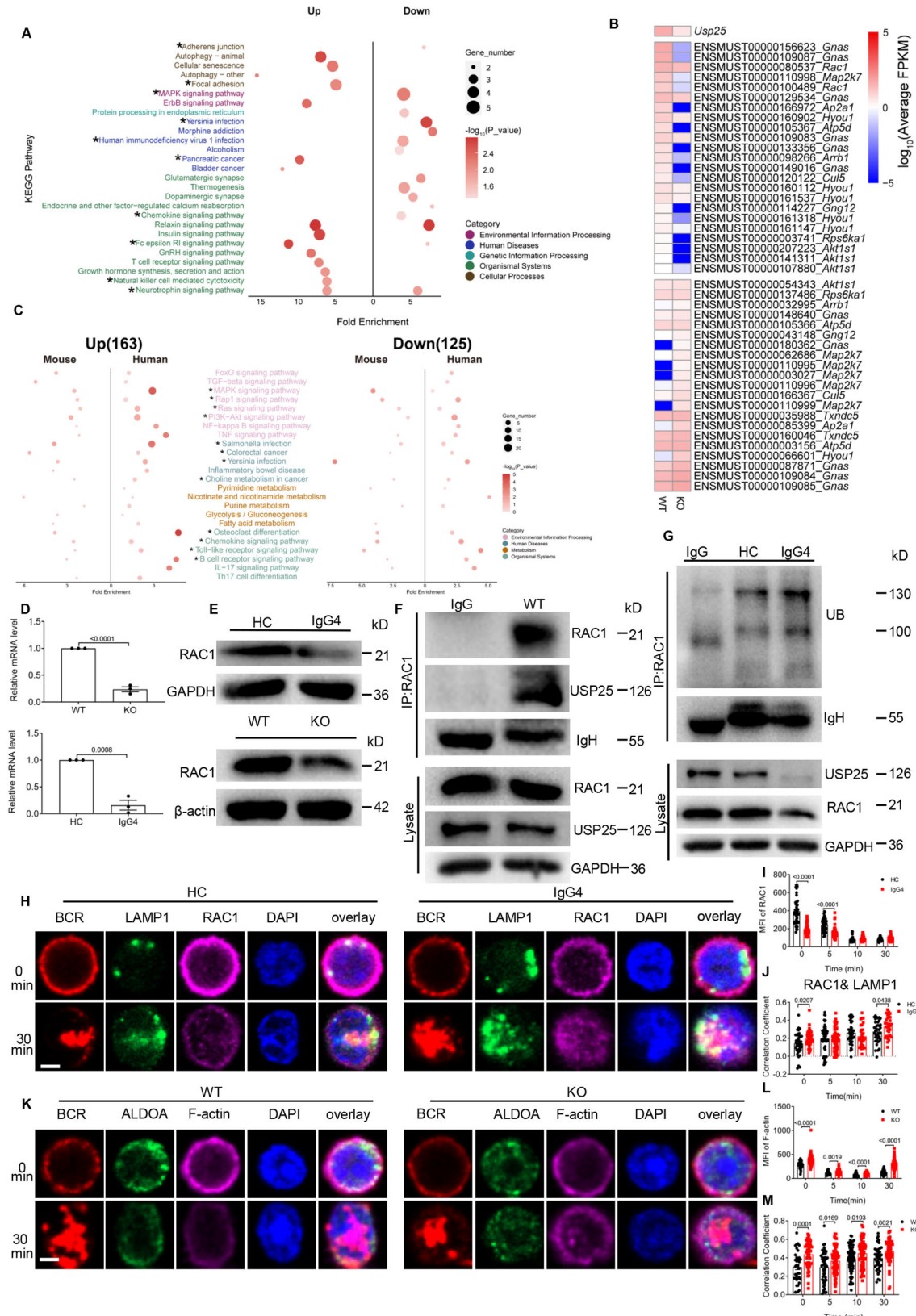

## Patients

We studied 43 diagnostic samples from 43 IgG4-RD patients (42–87 years old) at the Department of Rheumatology and Immunology, Wuhan Tongji Hospital (Supplementary Table 2). The clinical diagnosis of IgG4-RD has been made according to diagnostic criteria for IgG4-related disease[50]. Controls were healthy participants with matched gender and ages to the IgG4-RD patients. Informed consent was given by all participants according to CARE guidelines, MOST regulations and in compliance with the Declaration of Helsinki principles, and patient compensation was not provided. Due to the de-identified data, sex and gender information was not routinely collected and thus this analysis was not performed. This study was approved by the Ethics

**Fig. 6 | RAC1 participates in the regulation of glucose metabolism in B cells.**
**A** Top 15 enriched up-regulated and down-regulated KEGG pathways for DETs in *Usp25* KO mice models ($n = 3$). Asterisks represent the pathways associated with the RAC1 transcripts. **B** Heatmap showing the expression change of transcripts involved in the down-regulated KEGG pathways in mouse models. **C** KEGG pathway enrichment analysis of DETs in patients and mouse models. Asterisks represent the pathways associated with the *RAC1* gene ($n = 5$). **D, E** The mRNA levels (**D**) and protein (**E**) levels of HCs and IgG4-RD patients / *Usp25* KO and WT were analyzed by RT-PCR and Western blot, respectively ($n = 3$). **F** WT B cell lysates immunoprecipitated with RAC1 and immunoblotted for USP25. **G** HCs and IgG4-RD patients B cell lysates immunoprecipitated with RAC1 and immunoblotted for Ubiquitin.
**H–J** Purified B cells of IgG4-RD patients and HCs were stimulated in the same way as

in (Fig. 2G) except stained for LAMP1 and RAC1 (scale bar = 2.5 μm) and shown are representative images captured using confocal (**H**). The B cells were analyzed for the MFI of RAC1 (**I**) and the correlation coefficient of LAMP1/ RAC1 (**J**). **K–M** Purified B cells from *Usp25* KO and WT were stimulated with 10 μg/ml AF 594-F(ab′)₂ Ig (M + G), fixed, permeabilized, and stained for ALDOA, F-actin, and nuclei using DAPI (scale bar =2.5 μm) and shown are representative images captured using confocal (**K**). B cells were analyzed for the MFI of F-actin (**L**) and the correlation coefficient between F-actin and ALDOA (**M**). All images were representative images from 3 independent experiments. Data points in (**D, I, J, L**, and **M**) are represented as Mean ± SED. Statistical significance was based on two-tailed unpaired Student's t-test. Relevant p values are given in the graph. *$P < 0.05$; **$P < 0.01$; ***$P < 0.001$. Source data are provided as a Source Data file.

Committee of the Huazhong University of Science and Technology (reference TJ-IRB20220434).

## Animals

*Usp25* knockout (KO) mice in a C57BL/6 background have been described previously[17] and were kindly provided by Dr. Chen Dong (Institute for Immunology and School of Medicine, Tsinghua University, Beijing, China). The *Btk*-floxed mice in a C57BL/6 background have been described previously[51] and *Mb1*cre+/- mice were obtained from cyagen (C001056, China). *Btk*-floxed mice were crossed with *Mb1*cre mice to obtain BTK knockout mice (*Mb1*cre+/- *BTK* fl/fl). All mice were group-housed with five per cage under specific-pathogen-free (SPF) conditions with 12 h light/12 h dark cycle at 72 "F and 40% humidity with ad libitum access to food and water. Experimental and control mice were co-housed. Without special instruction, euthanasia and animal procedures were conducted on 6–8 weeks old male and/or female mice, age-matched and sex-matched. All experimental procedures with mice complied with the Chinese Council on Animal Care and were approved by the Ethics Committee of the Huazhong University of Science and Technology.

## Immortalization of B cells with Epstein-Barr virus

According to a previous method[52], PBMCs from IgG4-RD patients and HCs were infected with EBV and immortalized into B lymphoblastoid cell lines (B-LCL), which were cultured in RPMI1640 complete medium.

## Transfection of B cells

HEK293T cells (provided by Dr Hongmei Yang (Huazhong University of Science and Technology)) were inoculated in 10 cm plates overnight to achieve a - 50% confluence. The following day, phage-6tag-USP25 or the empty vector (provided by Dr Bo Zhong (Wuhan University)) was co-transfected with the packaging vectors pMD2G and pSPAX2 into HEK293T cells. The supernatant of the harvested virus was obtained to infect the HC cell line and IgG4-RD cell line after 48 h and the cells were collected for immunoblotting assays after 44 h.

## Cell collection, isolation and purification

For human blood samples, blood samples were collected into ethylenediamine tetraacetic acid (EDTA) tubes and processed in a biosafety level 2+ laboratory within 2 h post–blood draw. Whole blood was centrifuged at 300 g for 10 min at room temperature to collect the plasma. The remaining blood was diluted with PBS in a 1:1 ratio and then added into the tube of ficoll solution in a 1:1 ratio and centrifuged at 269 g for 20 min. After centrifugation, PBMCs were collected according to different densities. PBMCs were purified B cells were obtained by CD19 magnetic beads according to the manufacturer's instructions. For mouse samples, splenic lymphocytes were isolated by ficoll. Purification of spleen B cells was performed as described previously[53]. Briefly, purified splenic B cells were removed from T-cells by anti-Thy1.2 mAb and guinea pig complement, followed by incubation at 37 °C for 1 h.

## Flow cytometry and cell sorting

Fresh PBMCs were stained within 2 h after isolation. Before antibody staining, Fc receptors were blocked with FcR blocking at 1:50 concentration for 10 min and all antibodies were titrated to obtain optimized concentration. For cell surface staining, PBMCs ($1 \times 10^6$) and splenic lymphocytes ($1 \times 10^6$) were stained with antibodies using optimized concentration of antibodies for 30 min at 4°C together with fixable viability Stain or 7-Amino-Actinomycin D (7-AAD) at a concentration of 1:20. For intracellular staining, refer to the previous protocol[53]. For phosflow analysis, splenic B cells ($1 \times 10^6$) and PBMCs ($1 \times 10^6$) were stained with anti-B220 and anti-CD19, respectively, followed by 10 μg/ml biotin-F(ab′)₂ anti-human Ig (M + G) and 10 μg/ml biotin-conjugated F(ab′)₂ anti-mouse Ig (M + G), respectively, for 30 min. Then 20 μg/ml of streptavidin was added for 10 min on ice and activated at various times at 37 °C. Finally, after fixation and permeabilization, the cells were stained with the antibodies, followed by Alexa Fluor 405 goat anti-rabbit IgG and Alexa Fluor 488 goat anti-mouse IgG. Samples were measured by Attune™ NxT flow cytometer (Thermo Fisher) and tracking beads were used to ensure stability signals in flow cytometry batches. The MFI and percentages of B cell subsets were analyzed using FlowJo software. For naive and memory B cells sorting, fresh PBMCs ($2 \times 10^7$) were surface stained within 2 h of isolation, resuspended at $1 \times 10^{7/mL}$ in 2% FBS plus PBS, and sorted on a BD FACS Aria SORP (BD, FACS Aria SORP). Gating strategies for cell sorting of naive and memory B cells were depicted in Fig. S1. For flow cytometry, antibodies were added 1 μl per $10^6$ cells. For more details regarding the antibodies, refer to Supplemental Tables 3 and 4.

## Primary B cell activation and culturing in vitro

Purified B cells ($2 \times 10^6$) were grouped into three groups and were stimulated with either: 10 μg/ml LPS, 10 μg /ml LPS plus (0.5 ng/ml) TGF-β, or 8 ng/ml IL-4 plus 10 μg/ml CD40. The LPS group was collected on the fourth day and surface stained using B220 antibodies. After fixing and permeabilizing, they were divided into two groups and stained with IgG2b or IgG3, and then analyzed by flow cytometry. The remaining two groups of cells were collected on the fifth day and surface stained using B220 antibodies. After fixing and permeabilizing, the LPS plus TGF-β group was stained with IgA, and the IL-4 plus CD40 group was stained for IgG1, and then analyzed by flow cytometry. For flow cytometry, antibodies were added 1 μl per 10^6 cells. For more details regarding the antibodies and reagents, refer to Supplemental Tables 3 and 4.

## Proliferation, apoptosis and differentiation

Proliferation and apoptosis experiments were performed according to a previous method[53]. Briefly, purified splenic B cells ($1 \times 10^6$) and PBMCs ($1 \times 10^6$) were incubated with 5 μM Celltrace Violet for 5 min at 37 °C, and then cells were stimulated with LPS (5 μg/ml) or CPG (10 μg/ml) for 72 h and F(ab′)₂ anti-mouse Ig (M + G) (5 μg/ml) for 96 h. Apoptosis was measured by Annexin V/PI kit. For differentiation analysis, purified splenic B cells ($5 \times 10^5$) from WT and *Usp25* KO were

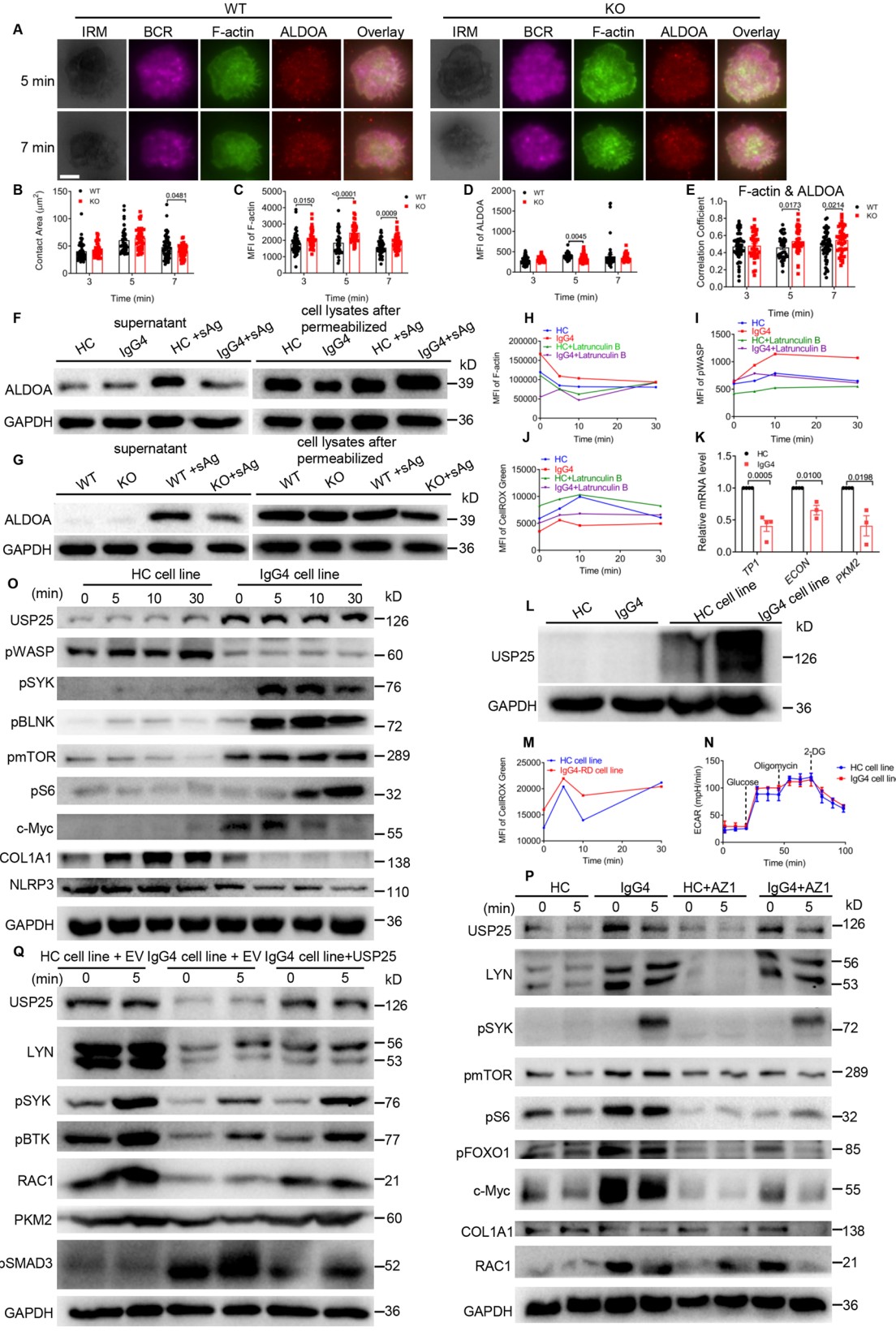

washed and resuspended in 10% FBS plus complete RPMI with 10 μg/ml LPS and incubated at 37 °C. Following 3 days of culturing, cells were marked with B220, CD138, and IgG1, and then fixed and analyzed by flow cytometer. For flow cytometry, antibodies were added 1 μl per 10^6 cells. For more details regarding the antibodies and reagents, refer to Supplemental Tables 3 and 4.

## ROS and Mito detection

For ROS analysis, purified splenic B cells (5 × 10^5) and PBMCs (5 × 10^5) were incubated with 10 μg/ml biotin-F(ab')$_2$ Ig (M + G) for 30 min and then 20 μg/ml of streptavidin was added for 10 min on ice. Subsequently, the cells were activated various times at 37 °C, followed by staining with CD19, CellROX Greenand B220. CellROX Green was

**Fig. 7 | Overexpressing USP25 in IgG4-RD cell line rescues the clinical phenotype of IgG4-RD patients. A–E** TIRFm analysis of the MFI of F-actin, ALDOA and the correlation coefficient between F-actin and ALDOA (scale bar = 2.5 μm). **F, G** PBMCs and splenic B cells were pre-incubated with sAg for 24 h and then permeabilized with 30 μg/ml of digitonin for 5 min and the protein levels of ALDOA in the supernatant and lysates were analyzed. **H, I** PBMCs were pre-incubated with 10 μM of Latrunculin B for 2 h at 37 °C and then incubated with anti-CD19, followed by stimulation with sAg. PBMCs were stained with F-actin and pWASP after fixation and permeabilisation, and analyzed (n = 3). **J** PBMCs were pre-incubated in the same way as in (Fig. 7H) except stained for anti-CD19 and CellIROX Green and the MFI of CellIROX Green were analyzed (n = 3). **K** RT-PCR was used to analyze the mRNA levels of *TP1, ECON,* and *PKM2* in purified B cells (n = 3). **L** Western blot of USP25 in PBMCs and cell lines. **M** Cell lines were analyzed for the MFI of CellIROX Green in the same way as in (Fig. 5N). **N** ECAR detection of B cells from HC cell line and IgG4-

RD cell line (n = 3). **O** Western blot of USP25, pWASP, pSYK, pBLNK, pmTOR, pS6, c-MYC, COL1A1, and NLRP3 in B cells of HC cell line and IgG4-RD cell line stimulated with sAg for designated times. **P** HC cell line and IgG4-RD cell line were pre-incubated with 10 μM of AZ1 for 1 h at 37 °C and then stimulated with sAg for 5 min. The protein levels of USP25, LYN, pSYK, pmTOR, pS6, pFOXO1, c-MYC, COL1A1 and RAC1 were analyzed. **Q** After overexpression of phage-6tag-USP25 or the empty vector in IgG4-RD cell line or HC cell line by lentiviral transfection, the protein levels of USP25, pBTK, LYN, pSYK, RAC1, PKM2, and pSMAD3 were analyzed (EV=Empty Vector). All images were representative images from 3 independent experiments. Data points in (**B–E, K**) are represented as Mean ± SED. Statistical significance was based on two-tailed unpaired Student's t-test. Relevant p values are given in the graph. *$P < 0.05$; **$P < 0.01$; ***$P < 0.001$. Source data are provided as a Source Data file.

added at 37 °C for 20 min and the cells were analyzed by flow cytometry. For Mito analysis, PBMCs, and purified splenic B cells were pre-incubated with 10 μg/ml biotin-F(ab')₂ Ig(M + G) at 37 °C for 24 h before being labeled with CD19, MitoTracker Green FM, and B220. MitoTracker Green FM was added at 37 °C for 15 min and the cells were analyzed by flow cytometry. For flow cytometry, antibodies were added 1 μl per $10^6$ cells. For more details regarding the antibodies and reagents, refer to Supplemental Tables 3 and 4.

## Confocal

Confocal microscopy was performed as described in a previous method[53]. Briefly, purified B cells ($2 × 10^6$) were incubated with 10 μg/ml Alexa Fluor 594-F(ab')₂ Ig(M + G) at 4 °C for 30 min. Then the cells were stimulated at 37 °C for varying lengths of time. Cells were fixed and permeabilized, followed by staining. For observing the B cells' mitochondria and cell morphology live, purified B cells were pre-stimulated with biotin-F(ab')₂ Ig(M + G) (10 μg/ml) at 37 °C for 24 h, and then incubated with CD19 plus PK Mito Red (PBMCs) or B220 plus PK Mito Red (mouse B cells) at 37 °C for 15 min. The cells were imaged on a Nikon confocal microscope. For Immunofluorescence staining, antibodies were diluted as 1:200. For more details regarding the reagents, refer to Supplemental Tables 3 and 4.

## Total internal reflection fluorescence microscope

For TIRFm analysis, purified B cells were incubated on antigen-tethered lipid bilayers various times at 37 °C. Following activation for various time points, the cells ($2.4 × 10^6$) were then fixed and stained with antibodies. Images were taken by TIRFm. For Immunofluorescence staining, antibodies were diluted as 1:200. For more details regarding the antibodies, refer to Supplemental Tables 3 and 4.

## Western blot

PBMCs and purified splenic B cells ($3 × 10^6$) were stimulated with 10 μg/ml biotin-F(ab')₂ Ig (M + G) for 30 min and then incubated with 20 μg/ml streptavidin for 10 min on ice. Subsequently, the cells were activated various times at 37 °C and lysed by RIPA buffer, and then loading buffer was mixed into the solution and heated for 10 min at 95 °C. Proteins were separated using a 10% SDS-PAGE and then probed with antibodies. For treatment with MG-132 inhibitor in vitro, purified splenic B cells and PBMCs were pre-treated with 25 μM of MG-132 at 37 °C for 30 min before incubating with 10 μg/ml biotin-F(ab')₂ Ig (M + G). For immunoblot, antibodies were diluted as 1:1000. For more details regarding the antibodies, refer to Supplemental Table 3 and Supplemental Table 4.

## Co-immunoprecipitation (CO-IP)

Purified splenic B cells ($6 × 10^6$) and PBMCs ($6 × 10^6$) were activated for 0 and 5 min using the same method as for the western blot described above. Cells were lysed with RIPA lysis buffer. Lysates were incubated with 1 μg of the indicated antibody for 2 h, followed by incubation with

30 μl protein G Sepharose overnight. The sepharose beads were washed with PBS and the immunoprecipitates were eluted using SDS loading buffer and boiling for 10 min. The samples were analyzed by standard immunoblot procedures.

## Scanning electron microscopy (SEM)

Sterile coverslips were pre-treated with poly-d-lysine solution before incubating with 10 μg/ml biotin-F(ab')₂ Ig (M + G) at 37 °C for 3 h. Purified B cells ($2×10^6$) collected from HCs and IgG4-RD patients were added to the coverslips coated with 10 μg/ml biotin-F(ab')₂ Ig (M + G) and stimulated for 10 min at 37 °C. Next, cells were washed and fixed with 1 ml of 2.5% glutaraldehyde, gradually dehydrated with increasing ethanol concentrations, and then dried to the critical point with liquid carbon dioxide. Cells were then captured by an Ultra-high Resolution SEM (SU8010, HITACHI). The expansion of cellular filopodia was imaged using ImageJ software (Bethesda).

## Transmission electron microscopy (TEM)

B cells purified ($1 × 10^6$) from humans (HCs and IgG4-RD patients) and mice (WT and *UPS25* KO) were fixed in 2.5% glutaraldehyde for 2 h at 4 °C. Cells were fixed and stained with 2% OsO4 in cacodylate buffer for 1 h at 4 °C, and then gradually dehydrated with increasing concentrations of ethanol and embedded in Epon. Samples were sectioned using an ultramicrotome and then put on Formvar-coated slot grids, followed by a coating with lead citrate. Cells were captured by TEM (H-7000FA, HITACHI, Japan).

## Seahorse analysis

Purified B cells ($8 × 10^6$) were pre-stimulated in 24-well plates with 10 μg/ml biotin-F(ab')₂ Ig (M + G) for 24 h, and then cells were collected and seeded into seahorse cell plates and incubated at 37 °C for 1 h. For Oxygen Consumption Rate (OCR) testing, the probe plates were added at the corresponding dosing: 1.5 μM Oligomyciine, 1 mM fluoro-carbonylcyanide phenylhydrazone (FCCP) and 500 nM Rotenone plus 1 μM Antimycine. For Extracellular Acidification Rate (ECAR) testing, the probe plates were added to the corresponding dosing: 10 mM Glucose, 2 μM Oligomycin, 5 mM 2-Deoxy-D-glucose (2DG). Plates were then run on the Seahorse XF24 analyzer (XFe24, Agilent Technologies). For more details regarding the reagents, refer to Supplemental Table 4.

## Determination of glutamic and cysteine

Serum glutamic and cysteine contents were measured using a glutamic assay kit and cysteine assay kit, respectively. For more details regarding the reagents, refer to Supplemental Table 4.

## Immunohistochemistry

After paraffin sections were deparaffinized, rehydrated and antigen repaired, endogenous peroxidase was blocked in hydrogen peroxide for 25 min at RT. Sections were incubated with anti-CD19 at 4 °C and

HRP conjugated goat anti-rabbit IgG (H + L) was applied for signal amplifications for 50 min at RT. For visualization, staining was visualized using a DAB Kit. Finally, sections were stained with hematoxylin and observed on a microscope.

### ELISA

Serum levels of IL-1β and IL-13 were quantified using ELISA kits according to the manufacturer's instructions. For more details regarding the reagents, refer to supplemental Table S4. To detect antibody affinity maturation, 96-well plates were coated with 5 µg/ml NP2 or 5 µg/ml NP30 antigen overnight at 4 °C with 100 µl of coating buffer. After blocking, diluted serum was applied to a 96-well plate and incubated at 37 °C for 60 min. NP-specific IgG1 were captured with HRP conjugated Goat anti-mouse IgG1 (1:50,000 dilution) for 60 min at 37 °C. Following plate washing, the reaction was incubated with TMB substrate for 10 min at 37 °C and then stopped by 10% sulfuric acid. Absorbance was recorded at 450 nm using an INFINITE 200 PRO(TECAN) reader.

### Immunization and LCMV infection

Eight-week-old C57BL/6 WT and *Usp25* KO mice were injected intraperitoneally (i.p.) with 50 µg 4-Hydroxy-3-nitrophenylacetyl-Keyhole Limpet Hemocyanin precipitated in adjuvant. Mice were sacrificed two weeks later and splenic lymphocytes were analyzed by flow cytometry. After NP-KLH immunization, the serum antibody levels were analyzed by ELISA. Eight-week-old C57BL/6 WT and *Usp25* KO mice were given i.p. injection of PBS containing 4*10^5 PFU (in 200 µl) of LCMV-Armstrong 53b. Mice were sacrificed a week later and B cells were analyzed by flow cytometry. After LCMV infection, the serum antibody levels of IgG1 were analyzed by ELISA.

### Real-time PCR

According to a previous method[53], total RNA from purified human B cells (2×10^6) and mouse B cells (2×10^6) were isolated using an RNA Kit and were retro-transcribed to gDNA using an RT Reagent Kit, and analyzed for cDNA expression levels by SYBR Premix Ex Taq TM using a Real-Time PCR system. Finally, the Ct value was assessed for gene expression by the $2^{-\Delta\Delta ct}$ formula with the primer sequences. For more details regarding the primer sequences and reagents, refer to Supplemental Table 5.

### Mutation analysis

Genomic DNA was extracted from IgG4-RD patients and HCs according to the instructions of the QIAamp DNA Micro Kit and was PCR-amplified using the forward primer in BLNK: 5'-GTCTTGCCCA-CACTCCAACG-3' and the reverse primer in BLNK: 5'-GTCAGGCAGT-GATGGGCACTG-3'primer and the mutation sites were analyzed by direct sequencing of the PCR products.

### Transcriptome sequencing and bioinformatics analysis

RNA extraction, quantity and quality assessment were performed as described previously[54]. Briefly, total RNA was extracted with the HiPure Universal RNA Mini Kit. The concentration, purity and integrity of total RNA were determined by the NanoDrop 2000 and Agilent Bioanalyzer 2100 System RNA Nano 6000 Assay Kit, respectively. RNA-seq libraries were constructed using AHTS Universal V8 RNA-seq Library Prep Kit for Illumina following the manufacturer's instructions, and then sequenced on GenoLab M platform and Illumina novaseq 6000 with a 150-cycle paired-end high-output sequencing mode. After filtering the adapter, ploy-N and low-quality reads, the reads were aligned to the human reference genome and mouse reference genome. The mapped reads were subjected to StringTie to perform expression quantification. Differentially expressed genes/transcripts (DEGs/DETs) were identified by DESeq2 (p.adj DEGs<0.05/DETs<0.01; |log2(fold change) |≥1). KEGG analysis was conducted with the R package clusterProfiler[55].

### Single-cell RNA-sequencing data processing

Single-cell RNA-sequencing data obtained from Wu et al.[6]. We used CellRanger v7.1.0 to generate and align raw gene expression data to the human reference genome (GRCh38). Single-cell RNA-sequencing data processing was conducted with Seurat v4.3.0. We filtered out cells that met any of the following criteria: (1) >10% of transcripts mapping to mitochondrial genes, (2) fewer than 500 total unique transcripts, and (3) a unique gene count of more than 3500 genes. To harmonize data across different samples and preserve the biological signals therein, we employed the Seurat package (v.4.3.0) and the reciprocal Principal Component Analysis (RPCA) approach. The NormalizeData function was used to normalize the data. FindVariableFeatures function was used to calculate 2,000 features with high cell-to-cell variation. The RunPCA function was used to reduce the dimensionality of the datasets at default parameters on linear-transformation scaled data generated by the ScaleData function. FindNeighbors and FindClusters functions were used to perform nonlinear dimensional reduction via the RunUMAP function (dims = 1:30, resolution = 0.8). Utilizing known marker genes specific to B cells, we precisely identified and isolated B cell populations for further focused analysis. For a more in-depth exploration of B cell subsets, FindNeighbors and FindClusters functions were used to perform nonlinear dimensional reduction via the RunUMAP function (dims = 1:30, resolution = 2.0). The identity and characteristics of the B cell subsets were meticulously confirmed through the scrutiny of relevant marker genes[26].

### Metabolic analysis in serum

For metabolomic analysis, data were acquired using mass spectrometry (MS) and analyzed using Progenesis QI and BGI's metabolomics package metaX. Pathway enrichment and visualization of differentially expressed metabolites were performed using the variable importance (VIP) in the PLS-DA model, which incorporates fold change (FC) (≤0.8333 or≥1.2) and q-values (<0.05). In addition, metabolic pathway enrichment was based on the KEGG database.

### Statistical and reproducibility

Statistical analyses were based on two-tailed unpaired Student's t-test and wilcoxon rank sum test using Prism GraphPad Prism 8 Software. Data were presented as the standard error of the mean (SEM). The statistical significant difference was donated as *$p < 0.05$, ** $p < 0.01$, *** $p < 0.001$ **** $p < 0.0001$. The exact p values were shown in the figures. The n number for all experiments were listed in the figure legends. All results were reproducible and representative data were shown in the figures or supplementary files. Investigators were blinded to group allocation during data collection, image quantification, and data analysis. No statistical method was used to predetermine the sample size. Sample sizes were selected based on previous experience and published literature to detect meaningful biological differences (https://pubmed.ncbi.nlm.nih.gov/36401985/). No data were excluded from analyses. Source data are provided in this paper.

### Reporting summary

Further information on research design is available in the Nature Portfolio Reporting Summary linked to this article.

## Data availability

The mouse RNA-Seq data generated in this study have been deposited in Genome Sequence Archive (GSA) database under accession code CRA014878. The human RNA-Seq data generated in this study have been deposited in GSA database under accession code HRA006661. The Single-Cell RNA-sequencing data generated in this study have been deposited in the GSA database under accession code HRA001555. Uncropped western blots for data in main and Supplementary Figs. are provided in the Source data file. Source data are provided with this paper.

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

## Acknowledgements

This study was supported by the National Natural Science Foundation of China (32311530061) to CH.L., National Key Research and Development Program of China (2023YFC2306100) to B.Z., International scientific and technological innovation cooperation between governments from Ministry of Science and Technology of the People's Republic of China (2021YFE0108200) to CH.L. and R&D Program of Guangzhou Laboratory (SRPG22-006) to CH.L. We thank Wanli Liu Prof. (Tsinghua University, Beijing, China) for constructive suggestions during the revised manuscript.

## Author contributions

P.J. drafted the paper, performed the flow cytometry, western blot, and generated figures. Y.J. and S.Z. performed the TIRFm experiments. C.L. assisted with the paper. L.Y. performed the infection experiments. X.D. performed the Co-Immunoprecipitation experiments. L.L. performed the confocal experiments. S.C. and Y.Z. helped collect samples. J.L., X.Z., and XC.Z. performed RNA-seq analysis. J.Y. and W.Z. performed Single-Cell RNA-Seq analysis. H.M., F.G., J.L., H.U. Y.W., B.Z., and L.D. reviewed and revised the manuscript. CH.L. conceived the idea and designed the research.

## Competing interests

Xin.Zh., Xia.Zh., and J.L. were employed by GeneMind Biosciences Company Limited, Shenzhen, China. Heather Miller was employed by Cytek Biosciences, R&D Clinical Reagents, Fremont, CA, United States. The remaining authors declare no competing interests.

## Additional information

¹Department of Pathogen Biology, School of Basic Medicine, Tongji Medical College and State Key Laboratory for Diagnosis and Treatment of Severe Zoonotic Infectious Diseases, Huazhong University of Science and Technology, Wuhan 430030 Hubei, China. ²Department Immunology, School of Medicine, Yangtze University, Jingzhou 434000, China. ³Department of Rheumatology and Immunology, Tongji Hospital, Tongji Medical College, Huazhong University of Science and Technology, Wuhan 430000, China. ⁴Cytek Biosciences, R&D Clinical Reagents, Fremont, CA, USA. ⁵GeneMind Biosciences Company Limited, Shenzhen 518001, China. ⁶Department of Nephropathy, the First Affiliated Hospital of Anhui Medical University, Hefei, Anhui 230022, PR China; Center for Scientific Research of Anhui Medical University, Hefei, Anhui 230032, PR China. ⁷Department of Orthopedics, Qilu Hospital of Shandong University, Jinan, Shandong 250063, PR China. ⁸Department of Rheumatology, Peking Union Medical College Hospital, Chinese Academy of Medical Science & Peking Union Medical College, National Clinical Research Center for Dermatologic and Immunologic Diseases, State Key Laboratory of Complex Severe and Rare Diseases, Beijing 100730, China. ⁹Department of Gastrointestinal Surgery, Medical Research Institute, Frontier Science Center for Immunology and Metabolism, Zhongnan Hospital of Wuhan University, Wuhan University, Wuhan 430071, China. ¹⁰TaiKang Center for Life and Medical Sciences, Wuhan University, Wuhan 430071, China. ¹¹Department of Medicine, Nagahama City Hospital, Nagahama 949-1701, Japan. ✉e-mail: tjhdongll@163.com; chaohongliu80@126.com

