## [Peer Review File · Nature Communications]

Expression of USP25 associates with fibrosis, inflammation and metabolism changes in IgG4-related diseaseREVIEWER COMMENTS

Reviewer #1 (Remarks to the Author):

Detailed comments below to author. There is a huge amount of missing information about the fundamental finding.

In this manuscript from Jiang et al. the authors argue that decreased USP25 expression in IgG4-RD B cells derives many of the changes in the disease. While there are many solid and interesting pieces of data, and clearly a lot of competence in analyzing signal transduction outputs there are crucial gaps in absolutely critical data that considerably weakens the overall narrative and certainly leaves this reviewer uncertain of the primary underlying thesis of the manuscript and relevance of the many unrelated findings.

1. The authors suggest alterations in the expression of CD19 based on MFI and BAFF-R also based on MFI – the levels are reduced in most B cell populations in IgG4-RD (Figures 1D and 1F):
- at the minimum the supplemental data should show overall gating strategies for flow cytometry
-In the main figures show primary flow plots for at least 3-4 patients and 3-4 controls gated on B cell subsets with single color flow staining depicted for CD19 and separately for BAFF-R (and also for Figures J, K, N and O, so that the reader can look at single cells and the actual levels in stained B cells in different subsets

2. A major issue in the paper is the absence of a more robust depiction of RNA-seq data comparing total B cells in IgG4-RD and healthy controls. In Figures 1 E and 2 F there are histograms showing relative expression of two chosen transcripts. In Figure 2F there are 5 RNAs from IgG4-RD subjects compared with 5 RNAs from healthy controls. In figure 2B there is a heat map with just 10 genes depicted from a total of 3 patients and 3 controls.

The quality-control data for the RNA seq studies remains uncertain. I suggest the RNA seq from all 10 subjects be included (5 patients and 5 controls) and the top 50 upregulated DEGs and the top 50 downregulated DEGs be depicted. Yes, there will be some variability – especially when performing bulk RNA seq on B cells whose compositions varies greatly between individuals. I had hoped to see some robust transcriptome data at least in the supplementary figures but was unable to find it. The lack of solid transcriptomic evidence weakens this manuscript and its central premise considerably. If such data does not exist. I suggest major B cell subsets be compared transcriptomically, and in a robust manner.

3. The discussion makes broad claims about USP25 but explains little and makes no attempt to explain or even consider fundamental issues. Whys are USP levels reduced in IgG4-Rd? Is this just because there is enrichment of certain B cell subsets in disease that express less USP25? Is this alteration because of an inflammatory milieu? If that is the case, then surely this change will not be unique to IgG4-RD.

4. Immunization with alum among many other experimental conditions help drive IgG1 responses in the mouse. High IgG4 is found in beekeepers and after allergenic desensitization. High IgG4 levels are not synonymous with IgG4-related disease. The mouse data is independently interesting but taking the leap to IgG4-related disease is asking for a suspension of disbelief.

Reviewer #2 (Remarks to the Author):

The study by Jiang et al. investigates the relationship between ubiquitin-specific proteinase 25 and pathophysiological aspects of IgG4-related diseases (IgG4-RD), including the altered B-cell signaling, altered humoral immunity and fibrosis. The study approaches an important topic of IgG4-RD pathophysiology, and various state-of-the-art methods are employed to investigate the impact of USP25 on the immunity. The discussed topic is novel and of high clinical and scientific relevance. The manuscript encompasses a considerable amount of work, presented in a coherent way.

Nevertheless, there is number of aspects, limiting my enthusiasm for the presented data:

1. The number of patients and controls used in the study and in each particular experiment largely vary: what are the reasons for it and what were the reasons and procedures for data exclusion? Lack of transparency in this process may lead to a considerable bias.
2. In descriptions to figures, patient numbers are given as a range: instead, it should be clearly specified, how many patients are in each group for every plot.
3. Authors show that the intensity of CD19+ signal is different between the groups (Fig. 1D); simultaneously, they show differences within the CD19+ positive population (Fig. 1A): are these differences influenced by each other?
4. Figure 2L: it is suggested that treatment with the ubiquitination inhibitor MG132 restores LYN degradation, however, it is not evident from the western data, while only pAKT and to a small extent pFOXO1 are affected by MG132.
5. Figure 3: it is argued that USP25 deficiency leads to impaired humoral response, however, it remains intriguing whether the antibody levels are also altered.
6. Figure 3: in Figure S3A, it is shown that KO mice have higher percentages of PBC as compared to WT mice. However, in Figure 3A and 3F, the total number of PBC+ in KO mice appears to be multiple-fold lower in KO mice as compared to WT mice, could this be elaborated?
7. Patients are described as representative of IgG4-RD without any specifics regarding the exact diagnosis, disease activity at the moment of sample collection, therapy, etc: this information is relevant, especially considering the heterogeneity of some of the presented results.
8. Discussion would benefit from additional context from the literature regarding USP25 molecular characteristics and its role in other diseases, such as Alzheimer, the potential links between those and IgG4-RD.
9. Occasional typing and language mistakes, please have it checked by a professional editor.

Reviewer #3 (Remarks to the Author):

In this manuscript the authors describe a role for the deubiquitinating enzyme USP25 in IgG4-related diseases. Using a plethora of patients-derived and mice-derived models the authors described how down-regulation of USP25 mimics many of the molecular phenomena altered in IgG4-RD patients. Their observations are relevant not only as they uncover molecular mechanism behind IgG4-RD but also brings to the light a potential disease marker and pharmacological target. However, the manuscript should be revised in different aspects before being accepted. Mainly in its English and in some additional data that could allow a better understanding of their observations and conclusions.

Major points:

1. One of the main points concern the English used throughout the manuscript, unfortunately I find many problematic sentences along the manuscript. I strongly suggest to the authors check again the manuscript, analyze carefully the English used and correct everything is necessary to be corrected. In this way the manuscript will really reflect all the work the authors performed, which is quite remarkable, in a better way. I will not list all the sentences with grammar problems but I will give just two examples.
 - line 24-25: "that ubiquitin-specific protease 25 (USP25) plays a role in regulating multiple pathways involved in symptoms", are they sure symptoms is the correct word to use?
 - line 47-48: "It is reported that the differentiation of B cells in IgG4-RD patients has altered", is instead of has.

2. The authors used an over-expression approach to demonstrate the direct role of USP25 in the IgG4-RD phenotypes. This is a very important point as could demonstrate whether the phenotypes observed in the *usp25* KO mice are just correlation or instead reflect a direct role of USP25 in the disease. Although the authors compared the over-expressed IgG4-RD with the HC cell line they should show a direct comparison between the IgG4-RD derived cell line with the respective USP25 over-expression cell line. This experiment is critical because will show the effect of restoring USP25 expression relative to the disease condition, showing if USP25 over-expression is able to rescue the different phenotypes, described along the paper, to the levels of the HC control cell line. For instance, the levels of pBTK, LYN, pSYK, RAC1, PKM2, pSMAD3 among others.

Minor comments

1. The figures are cited at the end of very long sentences including a lot of different observations in one shoot. As the figure are not cited at the end of each single data but instead at the end of a group of data the reader has to check by his own which is the figure corresponding to each panel. This made very difficult follow the manuscript. I strongly suggest to the authors include a figure citation at the end of the description of each data or panel.

2. In figure 2C is not specified how many cells were analyzed to generate the quantification and statistic analysis. This should be included.

3. In the different co-immunoprecipitation experiments were not included the immunoblot corresponding to the bait in the IP fraction. For instance, anti-LYN in figure 2H and anti-RAC1 in figure 6F. Showing IgG blot is not sufficient, instead it is necessary show how looks like the bait protein in the IP fraction as this reflects the efficiency of the IP.

5. Line 164-167. The authors described a mutation in the BLNK gene without explaining the meaning of that mutation. In other words, they don't provide a conclusion of explanation of the data, and so the reader has to interpretate by his own. Please conclude the sentence.

6. By analyzing LYN protein levels after B cells stimulation the authors conclude: 'we found that the degradation of LYN was accelerated upon stimulation in B cells with *Usp25* KO, indicating there is more ubiquitination of LYN'. This conclusion needs to be evaluated carefully. First, is clear that the protein levels of LYN are lower at time 5min in the KO compared to the WT but the protein levels were already lower at time zero. To be able to conclude about accelerated degradation the authors should quantified the percentage of LYN protein at time 5 relative to time zero, in both WT and KO cells, at least by triplicate for statistical porpoise. The so call accelerated degradation per se does not indicated more ubiquitination, this is more a speculation as they are independent observations, please correct the sentence.

7. The authors mention MG132 treatment restores USP25 levels in PBMCs. Please provide a quantitative analysis of this restoration as is not clear by only looking at the blot (figure 2L), otherwise limit the interpretation and eliminate this sentence. If the quantification confirms this conclusion please comment its significance, for instance is USP25 highly ubiquitinated and degraded in IgG4-RD samples?

8. MG132 is not a ubiquitination inhibitor (line 183), instead is a proteasome inhibitor. That's why stabilize ubiquitinated proteins. Please correct this sentence.

9. The blot signal of p-SMAD (figure 4H) is higher in IgG4-RD samples compared to HC samples in all the time points analyzed. Instead in figure 4G this increase was only scored at 5 min. Why the two analysis give different results? Please clarify.

10. The authors suggest USP25 regulated RAC1 ubiquitination and thus inhibiting RAC1 lysosome targeting and degradation. However, the authors are not showing data supporting the participation of the lysosome in RAC1 degradation in the IgG4-RD biological context, the differences scored in LAMP1 co-localization are not strong and sufficient to argue this point. This sentence should be eliminated or proved by analyzing the levels of RAC1 after lysosome activity inhibition.

11. RAC1 accelerated ubiquitination in IgG4-RD patients explain its down-regulation at the protein

level. But what about RAC1 gene expression down-regulation? Is because a feedback loop? This should be explained or at least suggest an explanation.

12. What reduces USP25 expression in IgG4-RD? The authors could provide one or more hypothesis in the discussion.

REVIEWER COMMENTS

Response to Reviewer # 1:

In this manuscript from Jiang et al. the authors argue that decreased USP25 expression in IgG4-RD B cells derives many of the changes in the disease. While there are many solid and interesting pieces of data, and clearly a lot of competence in analyzing signal transduction outputs there are crucial gaps in absolutely critical data that considerably weakens the overall narrative and certainly leaves this reviewer uncertain of the primary underlying thesis of the manuscript and relevance of the many unrelated findings.

Our response: We greatly appreciate this kind appraisal by Reviewer 1, and the thoughtful comments below which have significantly strengthened the revised manuscript. We have carefully addressed all comments in the revised manuscript, as detailed below.

1. The authors suggest alterations in the expression of CD19 based on MFI and BAFF-R also based on MFI – the levels are reduced in most B cell populations in IgG4-RD (Figures 1D and 1F):

- at the minimum the supplemental data should show overall gating strategies for flow cytometry

Our response: Thanks a lot for the great suggestions and details. In this new revised version and following the reviewer's advice, we have now included the overall gating strategies for flow cytometry data (please see Figure S1A).

-In the main figures show primary flow plots for at least 3-4 patients and 3-4 controls gated on B cell subsets with single color flow staining depicted for CD19 and separately for BAFF-R (and also for Figures J, K, N and O, so that the reader can look at single cells and the actual levels in stained B cells in different subsets

Our response: Thanks for these important suggestions and details. Due to the limited space in the figure, we have included the results of CD19 and BAFF-R single color flow staining in B cell subsets from 3 IgG4-RD patients and 3 HCs in the supplementary Figures S1B and S1C, respectively. The MFI of pY (please see Figure 1J), pBTK (please see Figure 1K), F-actin (please see Figure 1N) and pWASP (please see Figure 1O) were all analyzed using total internal reflection fluorescence microscopy (TIRFm) (please see Figure 1I&M). And we have clarified them in the Figure 1 legend (J-K, N-O).

2. A major issue in the paper is the absence of a more robust depiction of RNA-seq data comparing total B cells in IgG4-RD and healthy controls. In Figures 1

E and 2 F there are histograms showing relative expression of two chosen transcripts. In Figure 2F there are 5 RNAs from IgG4-RD subjects compared with 5 RNAs from healthy controls. In figure 2B there is a heat map with just 10 genes depicted from a total of 3 patients and 3 controls.

The quality-control data for the RNA seq studies remains uncertain. I suggest the RNA seq from all 10 subjects be included (5 patients and 5 controls) and the top 50 upregulated DEGs and the top 50 downregulated DEGs be depicted. Yes, there will be some variability – especially when performing bulk RNA seq on B cells whose compositions varies greatly between individuals. I had hoped to see some robust transcriptome data at least in the supplementary figures but was unable to find it. The lack of solid transcriptomic evidence weakens this manuscript and its central premise considerably. If such data does not exist. I suggest major B cell subsets be compared transcriptomically, and in a robust manner.

Our response: Thanks for this important suggestion. To confirm this fact in the case of the present study, and following the reviewer's advice, we now present a transcriptome analysis of B cells isolated from 5 HCs and 5 IgG4-RD patients. Firstly, we noticed that KEGG enrichment networks of both differential genes and transcripts indicated that the IL-17 signalling pathway may have a critical role in IgG4-RD pathogenesis (please see Figure 2A and Figure S2H-I), consistent with our previous data. Secondly, a transcript of *USP25* was significantly (\log_2 Fold Change/fold change= -11.63, $\text{padj}<0.01$) down-regulated and *USP25* was only involved in the IL-17 signalling pathway (please see Figure 2B and Figure S2I). Due to the lesser number of DETs, we showed all DETs in the IgG4-RD and HC samples by heatmap. The heatmap showed that 611 (388 up-regulated, 223 down-regulated) significant DETs were identified between IgG4-RD and HC samples ($|\log_2\text{Fold Chang}|\geq 1$, $\text{padj} < 0.01$, please see Figure S2J). Thus, the RNA seq data in the manuscript revision now includes 10 samples (5 IgG4-RD patients and 5 HCs), confirming that *USP25* plays a central role in IgG4-RD patients.

3. The discussion makes broad claims about *USP25* but explains little and makes no attempt to explain or even consider fundamental issues. Why are *USP* levels reduced in IgG4-Rd? Is this just because there is enrichment of certain B cell subsets in disease that express less *USP25*? Is this alteration because of an inflammatory milieu? If that is the case, then surely this change will not be unique to IgG4-RD.

Our response: We genuinely appreciate your insightful comment. *USPs*, as *DUBs*, regulate the immune response and inflammation, of which abnormal expression usually leads to a series of diseases, including cancer, brain diseases, neurodegenerative diseases, and infectious diseases (Shen J, et al. *Cell Prolif.* 2023;56(8): e13444.). In our study, *USP25* expression was significantly reduced in both purified B cells (please see Figure 2E) and PBMCs

(please see Figure 2G) from IgG4-RD patients. IgG4-RD is an autoimmune fibro-inflammatory disease and it is difficult to detect USP25 expression of IgG4-RD in a non-inflammatory state. Sjögren's syndrome (SS) is a systemic autoimmune disorder characterized by the chronic inflammation and dysfunction of exocrine glands (Baldini, et al. *Autoimmunity reviews*, 103425. 25 Aug. 2023,). However, we found increased USP25 levels in SS patients compared to HCs (unpublished data), implying that the reduced USP25 levels in IgG4-RD patients were not due to inflammation.

Investigating the factors that lead to reduced USP25 expression in IgG4-RD is indeed a crucial aspect of understanding the underlying mechanisms. IgG4-RD is a distinct systemic autoimmune-mediated disease. So far, numerous studies have provided evidence that epigenetic mechanisms are found to be related to the risk of various autoimmune diseases. DNA methylation is a well-studied epigenetic modification involved in gene expression regulation. (Jones PA. *Nat Rev Genet*. 2012;13(7):484–492.) Xunyao Wu et al. found aberrant DNA methylation modifications in peripheral B cells of IgG4-RD patients by genome-wide DNA methylation, proving the potential role of DNA methylation changes in the pathogenesis of IgG4-RD (Wu X, et al. *Arthritis Res Ther*. 2023;25(1):4). Meanwhile, we found DNA methyltransferase-related or demethylation-related genes differentially expressed in B cells of IgG4-RD patients by RNA-seq, including MTRR (log2FoldChange 10.88, padj<0.01), METTL23 (log2FoldChange 1.92, padj<0.01), TET2 (log2FoldChange -8.50, padj<0.01), USP9X (log2FoldChange -1.52, padj<0.01), HEMK1 (log2FoldChange -7.87, padj<0.01). Thus, we speculate that the reduction of USP25 in IgG4-RD patients may be a result of changes in DNA methylation. Additionally, we have incorporated a relevant discussion in the revised manuscript (Line 453-462).

4. Immunization with alum among many other experimental conditions help drive IgG1 responses in the mouse. High IgG4 is found in beekeepers and after allergenic desensitization. High IgG4 levels are not synonymous with IgG4-related disease. The mouse data is independently interesting but taking the leap to IgG4-related disease is asking for a suspension of disbelief.

Our response: Thanks again for this important question and we agree that high IgG4 levels are not synonymous with IgG4-related disease (IgG4-RD). IgG4-RD is characterized by multiorgan inflammation, elevated serum IgG4 concentration, tissue infiltration by IgG4 plasma cells, and storiform fibrosis in various organs, including salivary glands, kidney, pancreas, lung, lymph nodes, bile duct, prostate, retroperitoneum, and lacrimal glands (Kamisawa, et al. *Lancet (London, England)* vol. 385,9976 (2015): 1460-71.). So far, the pathogenic mechanism of IgG4-RD has not been fully discovered, possibly because of a lack of animal models that can fully mimic human IgG4-RD. Several mouse models have been established that have symptoms similar to IgG4-RD. The *Tgfbr2^{fspKO}* mice spontaneously developed autoimmune pancreatitis (AIP), which is one manifestation of IgG4-related sclerosing

disease (Boomershine, et al. *Gut* vol. 58,9 (2009): 1267-74.). MuSK-immunized mice predominantly induced IgG1 production. (Ulusoy, et al. *Clinical immunology (Orlando, Fla.)* vol. 151,2 (2014): 155-63.). LatY^{136F} knock-in mouse was featured as inflammatory mononuclear cell infiltration and fibrosis in multi-organs such as salivary gland, and pancreas, as well as excellent responsibility to glucocorticoid therapy. Additionally, IgG1+ positive cells were observed in the SMGs, pancreas, lungs and kidneys (Yamada, et al. *PLoS one* vol. 13,6 e0198417. 14 Jun. 2018). Mouse IgG1 and IgE are induced by Th2 cytokines, and IgG1 in mice and IgG4 in humans are the only IgG subclass unable to bind to C1q (Mestas J, Hughes CC. *J Immunol.* 2004;172(5):2731-2738.). These results suggest that mouse IgG1 and human IgG4 have some similarities. Some reports have described that IgG4-RD patients show a Th2-predominant immunoreaction, resulting in increased IgE production. (Zen Y, et al. *Hepatology.* 2007;45(6):1538-1546. Nakano, et al. *Clinica chimica acta; international journal of clinical chemistry* vol. 531 (2022): 261-264.).

In our study, increased IgG1 (please see Figure 3A-J) and IgE (please see Figure S5A-B) were detected in splenic B cells of *Usp25* KO mice, which is similar to the increased IgE and IgG4 in IgG4-RD. Meanwhile, *Usp25* KO mice had marked fibrosis in the pancreas, kidney, and liver (please see Figure 4A-D). Additionally, we detected increased expression of several inflammatory factors in the serum of *Usp25* KO mice, and found significant lymphocytic infiltration in the lungs after LPS stimulation in vivo (please see Figure 4I-M), which is similar to the fibroinflammatory process characterized in IgG4-RD. Finally, this study demonstrates metabolic similarities between *Usp25* KO mice and IgG4-RD patients (please see Figure 5 and Figure S7). Overall, the phenotype of *USP25* KO mice is similar to that of IgG4-RD. However, there is an urgent need for a mouse model to study the pathogenesis of IgG4-RD, and this is the aim of our study.

Response to Reviewer # 2:

The study by Jiang et al. investigates the relationship between ubiquitin-specific proteinase 25 and pathophysiological aspects of IgG4-related diseases (IgG4-RD), including the altered B-cell signaling, altered humoral immunity and fibrosis. The study approaches an important topic of IgG4-RD pathophysiology, and various state-of-the-art methods are employed to investigate the impact of USP25 on the immunity. The discussed topic is novel and of high clinical and scientific relevance. The manuscript encompasses a considerable amount of work, presented in a coherent way.

Nevertheless, there is number of aspects, limiting my enthusiasm for the presented data:

Our response: We thank Reviewer 2 for the thorough review and helpful comments and suggestions for improving our manuscript. Reviewer 2 indicates,

quite rightly, some weak points. Now we have carefully addressed all of these comments in the revised manuscript, as detailed below.

1. The number of patients and controls used in the study and in each particular experiment largely vary: what are the reasons for it and what were the reasons and procedures for data exclusion? Lack of transparency in this process may lead to a considerable bias.

Our response: Thank you for highlighting it. IgG4-RD is a rare disease, resulting in a limited number of patient samples, and to ensure the cellular activity of the patient samples, we performed the experiments as soon as we collected the samples. On the one hand, it is difficult to perform multiple experiments on a single sample at the same time because of the limited number of samples and the different number of cells required for each experiment. For example, flow cytometry analysis of cell surface staining requires at least 2×10^5 PBMCs, such as Figure 1C. Flow cytometry analysis of cell intracellular staining requires at least 1×10^6 PBMCs, such as Figure 1H. Western blot analysis of protein expression requires at least 3×10^6 B cells, such as Figure 1L. RT-PCR analysis of mRNA levels requires at least 2×10^6 B cells, such as Figure 2F. Confocal experiments require at least 2×10^6 B cells, such as Figure 2C. TIRFm experiments require at least 2.4×10^6 B cells, such as Figure 1I. RNA-sequencing experiments require at least 2×10^6 B cells, such as Figure 2A.

On the other hand, due to the same staining channels, many markers cannot be stained simultaneously in flow cytometry staining. For example, when analyzing the MFI of CD19 (please see Figure 1D), BAFF-R (please see Figure 1F), annexin V (please see Figure 1G) and CD79a (please see Figure 1J), they could not be stained simultaneously by flow cytometry owing to the CD19 (FITC anti-human CD19, Biolegend, 302206), BAFF-R (FITC anti-human BAFF-R, Biolegend, 316904), annexin V (FITC anti-Annexin V, Biolegend, 640906) and CD79a (FITC anti-human CD79a, Biolegend, 333512) antibodies both being FITC channels. Therefore, we did different experiments with samples from different patients, not intentionally excluding the data.

2. In descriptions to figures, patient numbers are given as a range: instead, it should be clearly specified, how many patients are in each group for every plot.

Our response: We appreciate the criticism. In this new revised version and following the reviewer's advice, we have now added specified patient numbers in each group for every figure legend.

3. Authors show that the intensity of CD19+ signal is different between the groups (Fig. 1D); simultaneously, they show differences within the CD19+ positive population (Fig. 1A): are these differences influenced by each other?

Our response: Thanks for this important question. To verify the effect of CD19+

signal intensity on MBC and naïve, we analyzed the MFI of IgD and CD27 in CD19+. There were no significant differences in the MFI of IgD and CD27 in CD19+ in IgG4-RD patients compared to HCs. Meanwhile, flow cytometry showed that CD19 expression was not consecutive and had a clear separation between CD19+ and CD19- in IgG4-RD patients, so CD19+ signal intensity did not affect the gating of MBC and naïve.

4. Figure 2L: it is suggested that treatment with the ubiquitination inhibitor MG132 restores LYN degradation, however, it is not evident from the western data, while only pAKT and to a small extent pFOXO1 are affected by MG132.

Our response: Thanks again for these details and we have replaced it with a better one (please see Figure 2L).

5. Figure 3: it is argued that USP25 deficiency leads to impaired humoral response, however, it remains intriguing whether the antibody levels are also altered.

Our response: We are sorry not to have been clear on this. When we immunized mice with NP-KLH to analyze the T cell-dependent immune response, we found that the titer of NP-specific IgG1 was increased in *Usp25* KO mice compared with WT mice after immunization (please see Figure 3J). Following this advice, we also have included the analysis of antibody affinity maturation. The results show that *Usp25* KO mice have reduced antibody affinity maturation compared to WT mice, shown by a significantly lower ratio of NP2-binding antibodies to NP30-binding antibodies (please see Figure S4G) indicating that USP25 deficiency leads to impaired humoral immune response. Thanks for this important suggestion and we have incorporated relevant experimental data in the revised manuscript (Line 200-201).

6. Figure 3: in Figure S3A, it is shown that KO mice have higher percentages of PBC as compared to WT mice. However, in Figure 3A and 3F, the total number of PBC+ in KO mice appears to be multiple-fold lower in KO mice as compared to WT mice, could this be elaborated?

Our response: We genuinely appreciate your insightful comment. First of all, the levels of PBC are total and not for specific antigens. Second, the antigens used are also totally different. Figure S3A, Figure 3A and Figure 3F are the results of LPS stimulation in vitro, NP-KLH immunization and LCMV infection of total PBC (non-antigen-specific PBC), respectively. *Usp25* KO mice is a conventional knockout (KO). It is possible that different antigenic stimulations with the help of different cells lead to different proportions of activated B cells differentiating into PBCs.

Polyclonal B cell stimulation leads to proliferation and differentiation of

activated B cells into antibody-secreting plasma cells (PCs). Depending on the type of antigen, the response can be T cell-independent (TI) or T cell-dependent (TD). TD responses to protein-based antigens require the cooperation of CD4 T cells, while TI antigens can stimulate antibody production in the absence of cognate T cell help. (Lesinski, G B, and M A Westerink. *Journal of microbiological methods* vol. 47,2 (2001): 135-49.). NP-KLH, a classical TD antigen that elicits well-characterized B cell responses. Activation of B cells leads to differentiation of B cells into either short-lived plasmablasts or to development of germinal centers that ultimately generate both long-lived antibody-secreting cells and memory B cells (Brewer, et al. *Cellular immunology* vol. 310 (2016): 78-88.). LCMV is a noncytolytic virus, and acute infection by the LCMV Armstrong strain can be rapidly controlled within a week due to a robust T-cell-mediated immune response, particularly involving CD8 T cells (Tang, et al. *Cellular & molecular immunology* vol. 20,5 (2023): 512-524.). Residual LCMV antigen depots can prime CD4 T cells for B cell help probably in a FcγR-depend manner, thereby promoting B-cell differentiation into LCMV-specific PBCs. (Schweier, et al. *European journal of immunology* vol. 49,4 (2019): 626-637.). LPS belongs to the TI antigens and stimulates dual receptor signaling by bridging the B cell receptor and Toll-like receptor 4 (BCR/TLR4). LPS is not only a potent stimulus for mouse B cell proliferation, but also for the generation of plasmablasts/plasma cells (PBs/PCs), being a model of choice for studying plasmocyte differentiation, in vitro (Björklund M, Coutinho A. *Eur J Immunol.* 1983;13(1):44-50.).

7. Patients are described as representative of IgG4-RD without any specifics regarding the exact diagnosis, disease activity at the moment of sample collection, therapy, etc: this information is relevant, especially considering the heterogeneity of some of the presented results.

Our response: Thank you for highlighting this. In the revised manuscript, following the reviewer's advice, we have collated basic clinical information on the IgG4-RD patients in Supplementary Table 1.

8. Discussion would benefit from additional context from the literature regarding USP25 molecular characteristics and its role in other diseases, such as Alzheimer, the potential links between those and IgG4-RD.

Our response: Thanks for this important suggestion. USP25 is a deubiquitinase that can regulate the progression of many diseases, including Alzheimer's disease (AD). AD is a progressive neurodegenerative disorder, neuropathology includes amyloid plaques, neurofibrillary tangles, synaptic dysfunction, and neuroinflammation in vulnerable brain regions. *USP25, a deubiquitinating enzyme encoded by chromosome 21*, is a critical regulator of AD pathology. USP25 overexpression resulted in defective neurogenesis, cognitive

impairments and increased amyloid deposition, which could contribute to the pathogenesis of Alzheimer's phenotypes (Zheng Q, et al. *Sci Adv.* 2021;7(1): eabe1340.). However, there are no symptoms associated with Alzheimer's disease currently observed in IgG4-RD patients.

9. Occasional typing and language mistakes, please have it checked by a professional editor.

Our response: We apologize for this oversight and we have edited language throughout to improve clarity by a native speaker.

Response to Reviewer # 3:

In this manuscript the authors describe a role for the deubiquitinating enzyme USP25 in IgG4-related diseases. Using a plethora of patients-derived and mice-derived models the authors described how down-regulation of USP25 mimics many of the molecular phenomena altered in IgG4-RD patients. Their observations are relevant not only as they uncover molecular mechanism behind IgG4-RD but also brings to the light a potential disease marker and pharmacological target. However, the manuscript should be revised in different aspects before being accepted. Mainly in its English and in some additional data that could allow a better understanding of their observations and conclusions.

Our response: We thank Reviewer 3 for the thorough review of our manuscript and very positive comments. We have carefully addressed all questions in the revised manuscript, as detailed below.

Major points:

1. One of the main points concern the English used throughout the manuscript, unfortunately I find many problematic sentences along the manuscript. I strongly suggest to the authors check again the manuscript, analyze carefully the English used and correct everything is necessary to be corrected. In this way the manuscript will really reflect all the work the authors performed, which is quite remarkable, in a better way. I will not list all the sentences with grammar problems but I will give just two examples.

- line 24-25: "that ubiquitin-specific protease 25 (USP25) plays a role in regulating multiple pathways involved in symptoms", are they sure symptoms is the correct word to use?

Our response: We apologize for this oversight. In the revised version we have replaced "symptoms" by "phenotypes" in the main text (Line 25-26).

- line 47-48: "It is reported that the differentiation of B cells in IgG4-RD patients has altered", is instead of has.

Our response: We apologize for this oversight. In the revised version we have replaced “has” by “is” in the main text (Line 50) and we have edited language throughout to improve clarity by a native speaker.

2. The authors used an over-expression approach to demonstrate the direct role of USP25 in the IgG4-RD phenotypes. This is a very important point as could demonstrate whether the phenotypes observed in the *usp25* KO mice are just correlation or instead reflect a direct role of USP25 in the disease. Although the authors compared the over-expressed IgG4-RD with the HC cell line they should show a direct comparison between the IgG4-RD derived cell line with the respective USP25 over-expression cell line. This experiment is critical because will show the effect of restoring USP25 expression relative to the disease condition, showing if USP25 over-expression is able to rescue the different phenotypes, described along the paper, to the levels of the HC control cell line. For instance, the levels of pBTK, LYN, pSYK, RAC1, PKM2, pSMAD3 among others.

Our response: Thanks for the great constructive suggestions. We overexpressed USP25 in IgG4-RD cell line, which have reduced USP25 expression, by lentiviral transfection of phage-6tag-USP25 or the empty vector, followed by western blot to detect that the protein levels of pBTK, LYN, pSYK, RAC1, PKM2, and pSMAD3 were all rescued to the levels of HC cell line (please see Figure 7Q). We believe that the great constructive suggestions significantly strengthen our manuscript and we have incorporated a relevant experimental data in the revised manuscript (Line 380-383).

Minor comments

1. The figures are cited at the end of very long sentences including a lot of different observations in one shoot. As the figure are not cited at the end of each single data but instead at the end of a group of data the reader has to check by his own which is the figure corresponding to each panel. This made very difficult follow the manuscript. I strongly suggest to the authors include a figure citation at the end of the description of each data or panel.

Our response: Thanks for this important suggestions and details. In the revised manuscript, following the reviewer's advice, we have included a figure citation at the end of the description of each data.

2. In figure 2C is not specified how many cells were analyzed to generate the quantification and statistic analysis. This should be included.

Our response: Thanks again for these details and we have added specified cells in Figure 2C legend (Line 922-923).

3. In the different co-immunoprecipitation experiments were not included the immunoblot corresponding to the bait in the IP fraction. For instance, anti-LYN in figure 2H and anti-RAC1 in figure 6F. Showing IgG blot is not sufficient, instead it is necessary show how looks like the bait protein in the IP fraction as this reflects the efficiency of the IP.

Our response: Thanks again for these details. Following reviewer's advice, in the revised version we have included anti-Lyn immunoblot in new Figure 2H and anti-RAC1 immunoblot in new Figure 6F.

5. Line 164-167. The authors described a mutation in the BLNK gene without explaining the meaning of that mutation. In other words, they don't provide a conclusion or explanation of the data, and so the reader has to interpretate by his own. Please conclude the sentence.

Our response: Thank you very much for this very important remark. This mutation in IgG4-RD patients alters the secondary structure of BLNK mRNA as predicted by the RNA Folding Form tool (please see: <http://www.unafold.org/mfold/applications/rna-folding-form.php>). We speculate that BLNK synonymous mutations in IgG4-RD patients result in a reduced production of BLNK mRNA transcripts, which may be due to synonymous mutation-mediated alterations in the secondary structure of BLNK mRNA and we have now concluded the sentence in our revised manuscript (Line 165-169).

6. By analyzing LYN protein levels after B cells stimulation the authors conclude: 'we found that the degradation of LYN was accelerated upon stimulation in B cells with Usp25 KO, indicating there is more ubiquitination of LYN'. This conclusion needs to be evaluated carefully. First, is clear that the protein levels of LYN are lower at time 5min in the KO compared to the WT but the protein levels were already lower at time zero. To be able to conclude about accelerated degradation the authors should quantified the percentage of LYN protein at time 5 relative to time zero, in both WT and KO cells, at least by triplicate for statistical porpoise. The so call accelerated degradation per se does not indicated more ubiquitination, this is more a speculation as they are independent observations, please correct the sentence.

Our response: Thanks a lot for pointing out this and we have corrected the sentence as suggested (Line 181-184).

7. The authors mention MG132 treatment restores USP25 levels in PBMCs. Please provide a quantitative analysis of this restoration as is not clear by only looking at the blot (figure 2L), otherwise limit the interpretation and eliminate this sentence. If the quantification confirms this conclusion please comment its

significance, for instance is USP25 highly ubiquitinated and degraded in IgG4-RD samples?

Our response: Thank you for highlighting this. Following the reviewer's advice, our quantitative analysis found that USP25 levels were elevated at 0 min after treatment of PBMCs from IgG4-RD patients with MG132 compared to PBMCs from untreated IgG4-RD patients, suggesting that USP25 is highly ubiquitinated and degraded in IgG4-RD patients (please see Figure S3I).

8. MG132 is not a ubiquitination inhibitor (line 183), instead is a proteasome inhibitor. That's why stabilize ubiquitinated proteins. Please correct this sentence.

Our response: We apologize for the error and we have now corrected it (Line 186).

9. The blot signal of p-SMAD (figure 4H) is higher in IgG4-RD samples compared to HC samples in all the time points analyzed. Instead in figure 4G this increase was only scored at 5 min. Why the two analysis give different results? Please clarify.

Our response: Thank you for highlighting this. The MFI of pSMAD3 (please see Figure 4G) is the result of confocal from Figure 4F, while pSMAD3 (please see Figure 4H) is the result of western blot. Although both confocal and western blot use the specific binding of antigen and antibody to detect the target protein, there are differences in methods. The proteins of western blot are denatured by heat to a linear structure, resulting in a change in conformation and function. Immunofluorescence fixes the cells to maintain the same morphology and structure as the original, and the fixed proteins are denatured and coagulated, which makes them different from the proteins in their natural state, but different from the linear structure of the western blot after denaturation.

10. The authors suggest USP25 regulated RAC1 ubiquitination and thus inhibiting RAC1 lysosome targeting and degradation. However, the authors are not showing data supporting the participation of the lysosome in RAC1 degradation in the IgG4-RD biological context, the differences scored in LAMP1 co-localization are not strong and sufficient to argue this point. This sentence should be eliminated or proved by analyzing the levels of RAC1 after lysosome activity inhibition.

Our response: Thank you for the reviewer's kind comment. Following the reviewer's advice, we found that the level of RAC1 in IgG4-RD patients indeed was rescued to some extent to that of HCs using lysosome activity inhibition, Bafilomycin A1 (please see Figures S8B-C). Thanks for this important

suggestion and we have incorporated relevant experimental data in the revised manuscript (Line 333-335).

11. RAC1 accelerated ubiquitination in IgG4-RD patients explain its down-regulation at the protein level. But what about RAC1 gene expression down-regulation? Is because a feedback loop? This should be explained or at least suggest an explanation.

Our response: Thank you for your valuable feedback and insightful suggestion. To verify whether there is feedback of decreased RAC1 protein level on RAC1 transcript level, we found a transcription factor, FMR1, which regulates RAC1 expression by TRRUST database (please see: <https://www.grnpedia.org/trrust/>). It was reported that the FMR1 protein can interact with CYFIP (cytoplasmic FMR1 interacting protein), a known downstream effector of RAC1, to modify FMR1 affinity for mRNA (Billuart P, Chelly J. *Neuron*. 2003;38(6):843-845. Napoli, et al. *Cell* vol. 134,6 (2008): 1042-54.) We speculated that the RAC1 protein may have a feedback regulation of RAC1 mRNA through FMR1 in IgG4-RD patients.

12. What reduces USP25 expression in IgG4-RD? The authors could provide one or more hypothesis in the discussion.

Our response: We genuinely appreciate your insightful comment. Investigating the factors that lead to reduced USP25 expression in IgG4-RD is indeed a crucial aspect of understanding the underlying mechanisms. In our revised manuscript, we have dedicated a section of the discussion to propose possible hypotheses regarding the reduction of USP25 expression in IgG4-RD (Line 453-462).

IgG4-RD is a distinct systemic autoimmune-mediated disease. So far, numerous studies have provided evidence that epigenetic mechanisms are found to be related to the risk of various autoimmune diseases. DNA methylation is a well-studied epigenetic modification involved in gene expression regulation. (Jones PA. *Nat Rev Genet*. 2012;13(7):484–492.). Wu X, et al. found aberrant DNA methylation modifications in peripheral B cells of IgG4-RD patients by genome-wide DNA methylation, proving the potential role of DNA methylation changes in the pathogenesis of IgG4-RD (Wu X, et al. *Arthritis Res Ther*. 2023;25(1):4. Published 2023 Jan 7.). Meanwhile, we found DNA methyltransferase-related or demethylation-related genes differentially expressed in peripheral B cells of IgG4-RD patients by RNA-seq, including MTRR (log2FoldChange 10.88, padj<0.01), METTL23 (log2FoldChange 1.92, padj<0.01), TET2 (log2FoldChange -8.50, padj<0.01), USP9X (log2FoldChange -1.52, padj<0.01), HEMK1 (log2FoldChange -7.87, padj<0.01). We speculate that the reduction of USP25 in IgG4-RD patients may be a result of changes in DNA methylation. We hope that our expanded discussion provides a more comprehensive exploration of this important aspect.

REVIEWER COMMENTS

Reviewer #1 (Remarks to the Author):

There are major issues with this manuscript have that not been addressed and the main concern of course it that some of the basic claims made are not supported by robust data. Even the seemingly very intriguing data on reduced USP25, especially on Western blot and concomitantly with Lyn and BCR activation linked phosphoproteins likely has a fairly trivial explanation. It could only potentially be an interesting direction if the obvious reason for this data had been carefully and rigorously tested and excluded but this was neither considered nor done. More comments below.

1. One major problem is the lack of robust, quality controlled transcriptomic data. The claim made in Figure 1 about CD19 RNA levels is not supported by a strong transcriptomic data set with multiple biological replicates and with a detailed heat map – so confidence is not engendered that this is solid data. This data was asked for and was not provided. Ideally tight subsets of B cells, say resting naïve B cells (IgD+CD27-CD21+CD11c-) or switched memory B cells (IgD-CD27+CD21+CXCR5+) should have been purified from healthy controls and IgG4-RD and quality controlled libraries prepared for RNAseq. The absolute need for this tight purification will be clear below, but there is not a single robust heat map that engenders any confidence in this manuscript.

2. The heat map in Figure 2B is a case in point. It is not a heat map that shows unbiased selection of the the top “up” or “down” genes. Any reader should have been able to observe at least the top 50 genes that go up and down in IgG4-RD B cells compared to the control counterpart B cells, but it would have been meaningful in each case ONLY if the cells were rigorously purified and carefully selected B cell subsets. They were not. On line 142 and the figure legend for Figure 2 both describe “transcriptomic profiles of B cells” from HCs and IgG4-RD patients. This is essentially a comparison of apples with oranges.

3. It is extremely well established (see PMID 37300833 for an example) that there are major changes in B cell subsets in IgG4-RD. The IgM memory/marginal zone B cells population is markedly diminished, Double negative B cells expand, “Activated naïve” B cells expand and so on. BCR activation with the anti G/M Fab2 reagent as well as steady state cells are likely to be very different. For even for the biochemical analyses that were done well in this manuscript – such as the Western blots and the phosphoprotein staining, the most obvious and trivial explanation for the differences seen is that very different cells are being compared.

4. The flow cytometry clearly does not appear to have been conducted rigorously. There are many reasons to be concerned about much of the flow cytometry data including the small MFI changes claimed, especially based on the authors’ own information and descriptions that were provided. MFI comparisons are extremely susceptible to batch effects and inadequate antibody optimization and it does take expertise to perform these well.

a) The transitional B cell gate is not at all resolved by CD24 and CD38 in Figure S1. It is hard to understand how one can be confident in analyzing transitional B cells using this data.

b) The lack of a viability stain being used in their supplemental gating strategy is troubling – it is not even mentioned in the Methods section. The authors go on to describe annexin V and PI proportions. That is problematic. The investigators should have gated out dead cells from their analyses before any flow cytometric analysis was performed.

c) Given the many MFI analyses, I assume the studies were done using freeze thawed PBMCs as batches on the same cytometer. Was the instrument's signal output kept consistent across experimental batches using rainbow tracking beads? Were the IgG4-RD samples run separate from healthy controls to dilute potential (likely) batch effects? There are no comments regarding this important element of MFI comparisons in the methods.

d) Regarding the CD19 MFI data - bar charts with error bars are really not acceptable – the authors should have shown the individual points for each PBMC

Technical issues and advice:

- a) No methodological data is provided regarding blood processing across IgG4-RD and healthy controls. Was the time between phlebotomy and PBMC isolation standardized? It should be mentioned in methods given the potential for cohort differences based on something unrelated to biology.
- b) Given the many MFI analyses, I assume the studies were done using freeze thawed PBMCs as batches on the same cytometer. Was the instrument's signal output kept consistent across experimental batches using rainbow tracking beads? Were the IgG4-RD samples run separate from healthy controls to dilute potential (likely) batch effects? There are no comments regarding this important element of MFI comparisons in the methods.
- c) Was each flow cytometry antibody titrated to ensure antigen saturation? No discussion of this in methods. The anti-CD19 antibody they used is one of the poorer ones on the market and the CD19 separation they show in the supplement is reasonable but not extremely well separated as CD19 stains often are or at least can be.

Some advice on Flow cytometry approaches for the investigators: Subtle differences in processing times, batch effects and cohort origin can account for small MFI changes. The distribution for each group needs to be shown (individual dot plots as has been done on other graphs), and indeed information that all the samples were processed at the same location and run on the cytometer at the same time. If controls and disease samples were run at different times, or if tracking beads were not used the data is invalid.

In future studies provide appropriate evidence that:

- a live dead exclusion dye was used (MFI differences cannot be accurately determined without this and if not used, the analysis is invalid)
- the same number of cells were stained for each sample to ensure antibody saturation of receptors
- control and disease cohorts were collected and processed in the same time period/same investigators
- samples were all run on the cytometer at the same time or rainbow tracking was to ensure the machine was calibrated between runs
- disease samples and controls were not run on separate days

Reviewer #2 (Remarks to the Author):

The authors did address comments in a thorough manner, which has improved the manuscript considerably.

Before it can proceed to the next steps, the relevant argumentation in the responses to the comments 1, 3 and 6 should find reflection in the manuscript.

Reviewer #3 (Remarks to the Author):

Thanks for addressing the comments. I recommend that the article be published.

Just a small correction to be done in line 331. LAMP1 decreases in IgG4-RD patients instead is writing increases.

Reviewer #1 (Remarks to the Author):

There are major issues with this manuscript have that not been addressed and the main concern of course it that some of the basic claims made are not supported by robust data. Even the seemingly very intriguing data on reduced USP25, especially on Western blot and concomitantly with Lyn and BCR activation linked phosphoproteins likely has a fairly trivial explanation. It could only potentially be an interesting direction if the obvious reason for this data had been carefully and rigorously tested and excluded but this was neither considered nor done. More comments below.

Our response: We thank Reviewer 1 for the thorough review and helpful comments and suggestions for improving our manuscript. Reviewer 1 indicates, quite rightly, some weak points. Now we have carefully addressed all of these comments in the revised manuscript and significantly improved the quality of the manuscript.

1. One major problem is the lack of robust, quality controlled transcriptomic data. The claim made in Figure 1 about CD19 RNA levels is not supported by a strong transcriptomic data set with multiple biological replicates and with a detailed heat map – so confidence is not engendered that this is solid data. This data was asked for and was not provided. Ideally tight subsets of B cells, say resting naïve B cells (IgD+CD27-CD21+CD11c-) or switched memory B cells (IgD-CD27+CD21+CXCR5+) should have been purified from healthy controls and IgG4-RD and quality controlled libraries prepared for RNAseq. The absolute need for this tight purification will be clear below, but there is not a single robust heat map that engenders any confidence in this manuscript.

Our response: We appreciate the constructive suggestions provided by the reviewer. In our initial analysis of bulk transcriptome sequencing, we did not observe any statistically significant differential changes in *CD19* expression. To address the reviewer's concerns regarding the bulk transcriptomics data, we followed the guidance of both the editor and reviewer by conducting single-cell transcriptomics on peripheral blood mononuclear cells (PBMCs) from healthy controls and IgG4-RD patients. While the *CD19* mRNA exhibited a downward trend in the single-cell transcriptomics data, this trend did not reach statistical significance (see A below). To bolster our findings, we carried out RT-PCR analysis on purified memory B cells and naïve B cells, revealing a notable reduction in *CD19* mRNA expression in IgG4-RD patients (please see Figure 1E). The potential reasons for the disparities between RT-PCR and RNA-Seq results may include factors such as sensitivity differences, RNA degradation, batch effects, biological heterogeneity, and more. In response to the previous review, the reviewer requested the inclusion of primary flow plots for a subset of patients and healthy controls, focusing on B cell subsets with single-color

flow staining for CD19. Due to space constraints in Figure 1, we have included the results of CD19 single-color flow staining in B cell subsets from 3 IgG4-RD patients and 3 healthy controls in the supplementary Figures S1B.

We strongly concur with the reviewer's viewpoint that conducting RNAseq analysis on sorted naive B cells and memory B cells is of utmost importance in studying the alterations in B cell subsets in IgG4-RD patients. This result is a crucial complement to our research. On one hand, cell sorting requires a sufficient number of highly active cells, while, on the other hand, fresh IgG4-RD samples are scarce, and the quantity of sorted memory B cells is too low to meet the sequencing starting amount or only allows for risk-prone library preparation. Since risk-prone library preparation cannot guarantee sequencing accuracy, and bulk RNAseq has lower sensitivity for low-abundance RNA. so, we have supplemented our results with 10x single-cell sequencing on peripheral blood mononuclear cells (PBMCs) from healthy controls and IgG4-RD patients. Remarkably, single-cell transcriptomics showed that *USP25* mRNA expression was significantly reduced in both naive B cells and memory B cells from IgG4-RD patients, which is consistent with the findings from bulk transcriptomics data (please see Figure 2B-E, S3A-E). It also showed that *USP25* mRNA expression was not affected by B cell subsets (please see Figure 2D). Furthermore, we confirmed the reduced *USP25* mRNA expression in purified memory B cells and naive B cells following sorting in IgG4-RD patients by RT-PCR (please see Figure 2F).

We are grateful for this valuable suggestion, and we have integrated the pertinent experimental data into the revised manuscript. This inclusion has not only strengthened our manuscript but has also bolstered the support for our conclusions.

A

Reviewers Figure 1: (A) The expression level of *CD19* on memory B cells and naive B cells in IgG4-RD patients and healthy controls was detected by single-

cell transcriptomics (n=3).

2. The heat map in Figure 2B is a case in point. It is not a heat map that shows unbiased selection of the the top “up” or “down” genes. Any reader should have been able to observe at least the top 50 genes that go up and down in IgG4-RD B cells compared to the control counterpart B cells, but it would have been meaningful in each case ONLY if the cells were rigorously purified and carefully selected B cell subsets. They were not. On line 142 and the figure legend for Figure 2 both describe “transcriptomic profiles of B cells” from HCs and IgG4-RD patients. This is essentially a comparison of apples with oranges.

Our response: We extend our gratitude to the reviewer for their insightful and constructive comments, and we wholeheartedly concur with the reviewer's viewpoint regarding the importance of describing the top 50 upregulated and downregulated differentially expressed genes (DEGs). However, our transcriptome analysis of B cells from 5 healthy controls and 5 IgG4-RD patients revealed a lesser number of differentially expressed transcripts, comprising 388 upregulated and 223 downregulated genes. Notably, the transcript of *USP25* was among the 63 downregulated DEGs. To ensure that readers can examine the top 50 genes that are differentially expressed in IgG4-RD patients B cells compared to healthy controls, we have included an Excel spreadsheet containing all DEGs in the revised Supplementary Material (please see Supplementary Table 3).

Our KEGG enrichment analysis indicated that the IL-17 signaling pathway may play a pivotal role in IgG4-RD pathogenesis (please see Figure 2A, S2H-I). Notably, *USP25* is significantly downregulated in the IL-17 signaling pathway and may be a key regulatory gene in this specific pathway (please see Figure S2I-K). In light of these findings, we conducted additional experiments to validate the conclusions derived from the bulk transcriptomics data. We have incorporated RT-PCR results (please see Figure 2F) into the manuscript. Furthermore, to bolster the evidence from our bulk transcriptomics data, we have integrated additional single-cell transcriptomics data on PBMCs from healthy controls and IgG4-RD patients, as suggested by the reviewer and editor. As mentioned, single-cell transcriptomics showed that *USP25* mRNA expression was significantly reduced in both naive B cells and memory B cells from IgG4-RD patients, which is consistent with the findings from bulk transcriptomics data (please see Figure 2B-E, S3A-E) and also confirmed through RT-PCR (please see Figure 2F).

We sincerely appreciate the reviewer's invaluable suggestions, which have significantly contributed to the enhancement of our manuscript.

3. It is extremely well established (see PMID 37300833 for an example) that there are major changes in B cell subsets in IgG4-RD. The IgM memory/marginal zone B cells population is markedly diminished, Double

negative B cells expand, "Activated naïve" B cells expand and so on. BCR activation with the anti G/M Fab2 reagent as well as steady state cells are likely to be very different. For even for the biochemical analyses that were done well in this manuscript – such as the Western blots and the phosphoprotein staining, the most obvious and trivial explanation for the differences seen is that very different cells are being compared.

Our response: Thanks for the constructive suggestions and we agree with the reviewer's observation that BCR activation with the anti-G/M Fab₂ reagent as well as steady state cells are likely to be very different. We definitely read lots of great papers from Shiv Pillai Professor and learned a lot before (please see references 3). Now we have carefully studied the article (Allard-Chamard H, Kaneko N, Bertocchi A, et al. Cell Rep. 2023;42(6):112630.), and then we have detected a reduced positive BCR signaling including pAKT by phosflow in naïve B cells and memory B cells of IgG4-RD patients, which is consistent with the results of western blot in the manuscript (see A-C below).

Reviewers Figure 2: (A) Gating strategy to examine B cell subsets by flow cytometry data. (B-C) PBMCs of IgG4-RD patients and healthy controls were stained with anti-CD19, anti-CD27 and anti-IgD antibodies for 30 min at 4°C together with fixable viability stain, followed by stimulation with 10 µg/ml biotin-F(ab')₂ Ig (M + G) plus 20 µg/ml streptavidin for 5 min at 37°C. Then fixed and permeabilized, cells were stained with anti-pATK and analyzed by flow cytometry. The MFI of pAKT was analyzed in naïve B cells and memory B cells of IgG4-RD patients by flow cytometry. Shown are representative plots (n=5).

4. The flow cytometry clearly does not appear to have been conducted

rigorously. There are many reasons to be concerned about much of the flow cytometry data including the small MFI changes claimed, especially based on the authors' own information and descriptions that were provided. MFI comparisons are extremely susceptible to batch effects and inadequate antibody optimization and it does take expertise to perform these well.

Our response: We greatly appreciate the reviewer's valuable suggestion. In the revised manuscript, following the reviewer's advice, we have carefully addressed all questions in the revised manuscript, as detailed below.

a) The transitional B cell gate is not at all resolved by CD24 and CD38 in Figure S1. It is hard to understand how one can be confident in analyzing transitional B cells using this data.

Our response: Thanks for these important suggestions and details. Currently, we use the conventional protocol of CD19⁺CD24^{high}CD38^{high} to classify transitional human B cells (Wirhth S, Lanzavecchia A. Eur J Immunol. 2005;35(12):3433-3441. Simon Q, Pers JO, Cornec D, Le Pottier L, Mageed RA, Hillion S. J Allergy Clin Immunol. 2016;137(5):1577-1584.e10.). Meanwhile, we have carefully studied and learned a lot from Shiv Pillai's and Ignacio Sanz's article on the classification of human B cell subsets (Allard-Chamard H, Kaneko N, Bertocchi A, et al. Cell Rep. 2023;42(6):112630. Kaminski DA, Wei C, Qian Y, Rosenberg AF, Sanz I. 2012; 3:302.). In the future, we will definitely consider using the method mentioned in the above two papers.

b) The lack of a viability stain being used in their supplemental gating strategy is troubling – it is not even mentioned in the Methods section. The authors go on to describe annexin V and PI proportions. That is problematic. The investigators should have gated out dead cells from their analyses before any flow cytometric analysis was performed.

Our response: We are sorry not to have been clear on this. Thanks for the reminder and we really appreciate the helpful and detailed suggestions on how to improve the paper. Definitely, dead cells should be gated out from analyses before any flow cytometric analysis is performed. In fact, all samples for flow cytometry in our manuscript were gated out dead cells before analysis were performed. However, anti-CD19 (FITC anti-human CD19, Biolegend, 302206), anti-BAFF-R (FITC anti-human BAFF-R, Biolegend,316904) and anti-CD79a (FITC anti-human CD79a, Biolegend, 333512) antibodies are conjugated with FITC and cannot be stained simultaneously. anti-CD19 (PerCP anti-human CD19 Antibody, Biolegend, 302228) and 7-AAD (BD Pharmingen, 559925) antibodies are conjugated with PerCP and cannot be stained simultaneously. Before analyzing the MFI of BAFF-R and CD79a, we detected cell viability using DAPI staining. The supplemental gating strategy

came from BAFF-R staining. In this new revised version and following the reviewer's advice, we have re-analyzed the MFI of BAFF-R and CD79a after purchasing the fixable viability stain 700 to label with dead cells, which is consistent with the previous results (please see Figure 1F, S1E&K-L). Also, we have replaced the supplemental gating strategy (please see Figure S1A) and included fixable viability stain in the methods (Line 543-544).

c) Given the many MFI analyses, I assume the studies were done using freeze thawed PBMCs as batches on the same cytometer. Was the instrument's signal output kept consistent across experimental batches using rainbow tracking beads? Were the IgG4-RD samples run separate from healthy controls to dilute potential (likely) batch effects? There are no comments regarding this important element of MFI comparisons in the methods.

Our response: We thank the reviewer for their insightful and constructive comment. To exclude batch effects, IgG4-RD samples and healthy controls samples were collected, stained and run in the same period. All samples were detected by Attune™ NxT sonic focused flow cytometer (Thermo Fisher). Before detection, we used performance tracking beads (Invitrogen, 2029773) to ensure consistent signals between flow cytometry batches. In this new revised version and following the reviewer's advice, we have now included the tracking beads (Line 552-553) and MFI descriptions (Line 553-554) in the methods.

d) Regarding the CD19 MFI data - bar charts with error bars are really not acceptable – the authors should have shown the individual points for each PBMC

Our response: Thanks for this important question. In this new revised version and following the reviewer's advice, we have replaced all bar charts with error bars with scatter dot plots.

Technical issues and advice:

a) No methodological data is provided regarding blood processing across IgG4-RD and healthy controls. Was the time between phlebotomy and PBMC isolation standardized? It should be mentioned in methods given the potential for cohort differences based on something unrelated to biology.

Our response: We thank the reviewer for pointing this out. In this new revised version and following the reviewer's advice, we have now added blood samples collection methods and PBMCs isolation time in the methods (Line 529-534).

b) Given the many MFI analyses, I assume the studies were done using freeze thawed PBMCs as batches on the same cytometer. Was the instrument's signal output kept consistent across experimental batches using rainbow tracking

beads? Were the IgG4-RD samples run separate from healthy controls to dilute potential (likely) batch effects? There are no comments regarding this important element of MFI comparisons in the methods.

Our response: We greatly appreciate the reviewer's valuable suggestion. As mentioned, we will definitely follow the reviewer's suggestion.

c) Was each flow cytometry antibody titrated to ensure antigen saturation? No discussion of this in methods. The anti-CD19 antibody they used is one of the poorer ones on the market and the CD19 separation they show in the supplement is reasonable but not extremely well separated as CD19 stains often are or at least can be.

Our response: We thank the reviewer for these constructive suggestions. In our manuscript, we titrated each of the flow cytometry antibodies and chose optimized concentration for cell staining. In this new revised version and following the reviewer's advice, we have now included a description of optimized concentration of antibodies in the methods (Line 540-544). The anti-CD19 antibody comes from Biolegend, which is the best antibody we currently have, and we will consider using other antibodies in the future.

Some advice on Flow cytometry approaches for the investigators: Subtle differences in processing times, batch effects and cohort origin can account for small MFI changes. The distribution for each group needs to be shown (individual dot plots as has been done on other graphs), and indeed information that all the samples were processed at the same location and run on the cytometer at the same time. If controls and disease samples were run at different times, or if tracking beads were not used the data is invalid.

Our response: Thanks a lot for the detailed advice and we have checked each data carefully and indeed did the experiments as suggested. In this new revised version and following the reviewer's advice, we have replaced all bar charts with error bars with scatter dot plots, added sample processing time, the MFI descriptions and use of tracking beads to our methods (Line 539-544, 551-554).

In future studies provide appropriate evidence that:

- a live dead exclusion dye was used (MFI differences cannot be accurately determined without this and if not used, the analysis is invalid)

Our response: We greatly appreciate the reviewer's valuable suggestion. As mentioned, we will definitely follow the reviewer's suggestion.

- the same number of cells were stained for each sample to ensure antibody saturation of receptors

Our response: Thanks for the reminder. In our manuscript, the same number of cells were stained for each sample. In this new revised version and following the reviewer's advice, we have now included a description of the number of cells for each sample in the methods.

- control and disease cohorts were collected and processed in the same time period/same investigators

Our response: Thank you for highlighting it. In our manuscript, peripheral blood samples (IgG4-RD and healthy controls) were collected and processed in the same period/ same investigators.

- samples were all run on the cytometer at the same time or rainbow tracking was to ensure the machine was calibrated between runs

Our response: Thanks for the reminder again. In our manuscript, all samples were detected by Attune™ NxT sonic focused flow cytometer (Thermo Fisher). Before detection, we used performance tracking beads (Invitrogen, 2029773) to ensure consistent signals between flow cytometry batches. In this new revised version and following the reviewer's advice, we have now included the tracking beads descriptions in the methods (Line 552-553).

- disease samples and controls were not run on separate days

Our response: We thank the reviewer for this important notice. IgG4-RD and healthy controls samples were collected, stained and detected in the same period. We would like to take this opportunity to thank you for all your time involved and this great opportunity for us to improve the manuscript. We hope you will find this revised version satisfactory.

Reviewer #2 (Remarks to the Author):

The authors did address comments in a thorough manner, which has improved the manuscript considerably.

Before it can proceed to the next steps, the relevant argumentation in the responses to the comments 1, 3 and 6 should find reflection in the manuscript.

Our response: Thank you for your valuable feedback and insightful suggestions. Following the reviewer's advice, in the revised version we have added comments 1 to the methods, comments 3 to the results (lines 96-99, Fig. S1C-D) and comments 6 to the discussion (Line 426-429).

Reviewer #3 (Remarks to the Author):

Thanks for addressing the comments. I recommend that the article be published.

Just a small correction to be done in line 331. LAMP1 decreases in IgG4-RD patients instead is writing increases.

Our response: Thank you for the reviewer's kind comment and we have now corrected it (Line 348).

REVIEWER COMMENTS

Reviewer #1 (Remarks to the Author):

This manuscript's data is fine, the reduction in USP25 could be interesting but the interpretation of the data overall is still quite problematic. The authors could easily fix this with some rewriting. The new data actually confirms what has been my obvious suspicion all along. While I am glad that single cell RNA seq data has been undertaken if the authors provide one piece of information (that they already, have) and correctly and cautiously re-interpret their results and accordingly are more cautious in the writing of the Title, abstract, results and Discussion then I am perfectly fine with publication. As described I am pretty sure what is provided is not the right interpretation and they would do themselves, the B cell field and the field of human autoimmune diseases a disservice by publishing with the current interpretations that they have.

In the literature and in work from many other groups, the human "naive B cell gate" when studied in any depth, contains six discernible B cell populations, and one population, so called "activated naive" B cells, expands greatly in disease contexts. Human switched/memory B cells contain an additional six populations at the minimum, including two populations that have been seen to expand in many diseases including lupus and IgG4-related disease.

The authors's limited single cell data lacks the depth to separate these populations either in the "naive" gate or the "memory" gate and they come up with just ONE "naive" B cell population and ONE memory B cell population in patients and in the controls. THEY MUST PROVIDE the heat map from the single cell data comparing control "naive" vs IgG4-RD "naive", and control "memory" vs IgG4-RD "memory". They have the data so this should be easy.

The obvious and simple interpretation of this data is that disease related B cells dominate in IgG4-RD in the "naive" gate and also in the memory gate, and these disease-related subsets express less USP25. There is no reason to believe that this finding has anything to do with IgG4-RD specifically but likely just reveals that activated B cells that expand in disease contexts express less USP25.

Reviewer #2 (Remarks to the Author):

No further comments, the manuscript can be considered for publication.

Reviewer #1 (Remarks to the Author):

This manuscript's data is fine, the reduction in USP25 could be interesting but the interpretation of the data overall is still quite problematic. The authors could easily fix this with some rewriting. The new data actually confirms what has been my obvious suspicion all along. While I am glad that single cell RNA seq data has been undertaken if the authors provide one piece of information (that they already, have) and correctly and cautiously re-interpret their results and accordingly are more cautious in the writing of the Title, abstract, results and Discussion then I am perfectly fine with publication. As described I am pretty sure what is provided is not the right interpretation and they would do themselves, the B cell field and the field of human autoimmune diseases a disservice by publishing with the current interpretations that they have.

In the literature and in work from many other groups, the human "naive B cell gate" when studied in any depth, contains six discernible B cell populations, and one population, so called "activated naive" B cells, expands greatly in disease contexts. Human switched/memory B cells contain an additional six populations at the minimum, including two populations that have been seen to expand in many diseases including lupus and IgG4-related disease.

The authors's limited single cell data lacks the depth to separate these populations either in the "naive" gate or the "memory" gate and they come up with just ONE "naive" B cell population and ONE memory B cell population in patients and in the controls. THEY MUST PROVIDE the heat map from the single cell data comparing control "naive" vs IgG4-RD "naive", and control "memory" vs IgG4-RD "memory". They have the data so this should be easy.

The obvious and simple interpretation of this data is that disease related B cells dominate in IgG4-RD in the "naive" gate and also in the memory gate, and these disease-related subsets express less USP25. There is no reason to believe that this finding has anything to do with IgG4-RD specifically but likely just reveals that activated B cells that expand in disease contexts express less USP25.

Our response: Thank you for your valuable feedback and insightful suggestions. We really appreciate it and it significantly improved the quality of the manuscript. In this new revised version and following the reviewer's advice, we have clustered B cell scRNA-seq data with higher resolution, and 11 B cell subsets were identified (please see Figure 2B-D) (Sanz I, Wei C, Jenks SA, et al. Front Immunol.2019;10:2458.). As expected, activated naive B cells (BN-A), identified by IgD+CD27- CD38-CD24-CD21-CD11c+T-bet+, nearly all of them were in IgG4-RD, and few of them were found in healthy controls, consistent with previous studies (Allard-Chamard H, Kaneko N, Bertocchi A, et al. Cell Rep.

2023;42(6):112630.). Age-associated B cells (ABC), identified by IgD⁻CD27⁻CD38⁻CD24⁻CD21⁻CXCR5⁻CD11c⁺, were found no difference between IgG4-RD and healthy controls, while previous studies have reported them expanded in IgG4-RD (Zhang P, Lu H, Peng Y, et al. Clin Exp Rheumatol. Published online July 13, 2023.). This might be due to the low cell counts of ABC in scRNA-seq data and the small sample size of scRNA-seq data. Activated switched memory B cells (BM-SA), identified by IgD⁻CD27⁺CD38⁻CD24⁻CD21⁻CD86⁺IgG/IgA⁺, were expanded obviously in IgG4-RD (please see Figure S3E-F). Since scRNA-seq is low depth sequencing, it is not always possible for scRNA-seq data to fully annotate all cell populations, especially all six naive B cell populations and six memory B cell populations, which depends on the sequencing depth, cell counts, and clustering results. In fact, our annotation results included 5 naive B cell subsets and 5 memory B cell subsets with high resolution. To facilitate further analysis, some B cell subsets with less than 20 cells would be combined with other B cell subsets as described in the manuscript (Line 168-174). We have added the heat maps comparing naive B cell subsets between IgG4-RD and healthy controls, as well as memory B cell subsets, as suggested by the reviewer and we found that naive B cells showed more expression changes of DEGs than other B cell subsets (please see Figure S4A-G).

Furthermore, we have also checked the USP25 expression levels of each B cell subset. We found that in IgG4-RD, USP25 expression was significantly decreased in resting naive B cells (BN-R). Activated naive B (BN-A) cells were nearly all in IgG4-RD, and the USP25 expression level in them was lower than in resting naive B cells of healthy controls. In nearly all memory B cells, including pre-switched (BM-PS), resting switched (BM-SR), activated switched memory B cells (BM-SA), and ABC, USP25 expression levels in IgG4-RD were observed to be lower than healthy controls, albeit without statistical significance. In transitional type 1 B cells (BT1), transitional type 2 B cells (BT2), and unswitched memory B cells (BM-US), USP25 expressions in IgG4-RD were higher than healthy controls without statistical significance, but these B cell subsets in IgG4-RD were diminished (please see Figure 2E). Therefore in total, USP25 in B cells was expressed less.

We wholeheartedly concur with the reviewer's viewpoint regarding disease related B cells dominate in IgG4-RD in the "naive" gate and also in the memory gate, and these disease-related subsets express less USP25. In the revised manuscript, following the reviewer's advice, we have carefully and cautiously re-interpreted the Title, Abstract, Results, and Discussion. We sincerely appreciate the reviewer's invaluable suggestions, which have significantly contributed to the enhancement of our manuscript.

Reviewer #2 (Remarks to the Author):

No further comments, the manuscript can be considered for publication.

Our response: We greatly appreciate the reviewer's valuable suggestion.

REVIEWERS' COMMENTS

Reviewer #1 (Remarks to the Author):

While the manuscript is softened and I do not want to stand in the way of publication, the approach of comparing bulk B cells between any disease and controls for transcriptomics will always be fundamentally flawed in this day and age. Transcriptomic differences thus defined reflect expansions of activated B cell subsets and these types of transcriptomic changes will be seen not just in many different inflammatory diseases (beyond IgG4-RD) but even after immunization and infection.

REVIEWERS' COMMENTS

Reviewer #1 (Remarks to the Author):

While the manuscript is softened and I do not want to stand in the way of publication, the approach of comparing bulk B cells between any disease and controls for transcriptomics will always be fundamentally flawed in this day and age. Transcriptomic differences thus defined reflect expansions of activated B cell subsets and these types of transcriptomic changes will be seen not just in many different inflammatory diseases (beyond IgG4-RD) but even after immunization and infection.

Our response: Thank you for your valuable feedback and insightful suggestions. We really appreciate it and it significantly improved the quality of the manuscript. We have incorporated a relevant discussion in the revised manuscript (please see Line 447-451).